# Quantitative analysis of how Myc controls T cell proteomes and metabolic pathways during T cell activation

**Julia M Marchingo, Linda V Sinclair\*, Andrew JM Howden, Doreen A Cantrell\***

Cell Signalling and Immunology Division, School of Life Sciences, University of Dundee, Dundee, United Kingdom

**Abstract** T cell expansion and differentiation are critically dependent on the transcription factor c-Myc (Myc). Herein we use quantitative mass-spectrometry to reveal how Myc controls antigen receptor driven cell growth and proteome restructuring in murine T cells. Analysis of copy numbers per cell of >7000 proteins provides new understanding of the selective role of Myc in controlling the protein machinery that govern T cell fate. The data identify both Myc dependent and independent metabolic processes in immune activated T cells. We uncover that a primary function of Myc is to control expression of multiple amino acid transporters and that loss of a single Myc-controlled amino acid transporter effectively phenocopies the impact of Myc deletion. This study provides a comprehensive map of how Myc selectively shapes T cell phenotypes, revealing that Myc induction of amino acid transport is pivotal for subsequent bioenergetic and biosynthetic programs and licences T cell receptor driven proteome reprogramming.

**\*For correspondence:**
l.v.sinclair@dundee.ac.uk (LVS);
d.a.cantrell@dundee.ac.uk (DAC)

**Competing interests:** The authors declare that no competing interests exist.

## Introduction

Immune activation transcriptionally reprograms T lymphocytes and initiates changes in cell metabolism and protein synthesis that are required for proliferation and effector differentiation.

The signalling pathways that control T cell metabolism are not fully characterised but it has been shown that the transcription factor Myc has a necessary role (*Wang et al., 2011*). In T cells, Myc is rapidly induced in response to engagement of the T cell antigen receptor (TCR) and Myc expression is then sustained by costimulatory receptors and cytokines such as interleukin-2 (IL-2) (*Au-Yeung et al., 2017*; *Heinzel et al., 2017*; *Preston et al., 2015*). The TCR acts as a digital switch for *Myc* mRNA expression, in that the strength of the antigen stimulus determines the frequency of T cells that switch on *Myc* mRNA expression (*Preston et al., 2015*). Antigen receptor, costimulation and cytokine driven processes also post-transcriptionally control Myc protein: constant phosphorylation on Thr58 by glycogen synthase kinase 3 (GSK3) and subsequent proteasomal degradation results in a short cellular half-life of Myc protein (*Preston et al., 2015*). O-GlcNAcylation of Myc at this same residue (*Chou et al., 1995*), fuelled by the hexosamine biosynthesis pathway, blocks this degradation and allows Myc to accumulate (*Swamy et al., 2016*). In activated lymphocytes the sustained expression of Myc is also dependent on the rate of protein synthesis and availability of amino acids (*Loftus et al., 2018*; *Sinclair et al., 2013*; *Swamy et al., 2016*; *Verbist et al., 2016*). Myc expression is thus tightly controlled at the population and single cell level during immune responses.

The expression of Myc is essential for T cell immune responses and mature T cells with *Myc* alleles deleted cannot respond to antigen receptor engagement to proliferate and differentiate (*Preston et al., 2015*; *Trumpp et al., 2001*; *Wang et al., 2011*). Myc-deficient T cells have defects in glucose and glutamine metabolism (*Wang et al., 2011*); however, the full molecular details of how Myc regulates T cell metabolic pathways and other aspects of T cell function is not fully understood. In this context there are different models of how Myc works and divergent opinions as to

**eLife digest** T cells are white blood cells that form an important part of our immune defence, acting to attack disease-causing microbes and cancer and directing other immune cells to help in this fight. T cells spend most of their time in a resting state, small and inactive, but when an infection strikes, they transform into large, active 'effector' cells. This change involves a dramatic increase in protein production, accompanied by high energy demands. To fully activate, T cells need to boost their metabolism and take in extra amino acids, the building blocks of proteins. For this, they depend upon a protein called Myc.

The Myc protein works as a genetic switch, controlling several kinds of cell metabolism, but the molecular details of its effects in T cells remain unclear. Most studies looking to understand Myc have focussed on its role in cancer cells. Here its main job is thought to be driving the use of sugar to make energy. However, it has also been shown to control the levels of transporters that carry amino acids into cells and thus provide the raw materials for protein production. It is possible that Myc plays a similar role in T cells as it does in cancer cells, but this might not be the case because cancer cells have strange biology and do not always accurately represent healthy cells.

To find out what role Myc plays in T cell activation, Marchingo et al. compared T cells with and without Myc. The cells lacking Myc were much smaller than their normal counterparts and counts of their proteins revealed why. Without Myc, protein production had stalled. In normal T cells, the number of amino acid transporters increased up to 100 times as cells transformed from a resting to an active state. But, without Myc, this did not happen. The loss of Myc cut off the supply of amino acids, halting protein production. For T cells, the most important amino acid transporter is a protein called System-L transporter Slc7a5. It supplies several essential amino acids, including methionine – the amino acid that starts every single protein. To confirm the role of amino acid transporters in T cell activation, Marchingo et al. deleted the gene for the System-L transporter Slc7a5 directly. This had the same effect as deleting the gene for Myc itself, demonstrating that a key role of Myc in T cell activation is to increase the number of amino acid transporters.

Understanding the role of Myc in T cell activation is an important step towards controlling the immune system. At the moment, many research groups are investigating how best to use T cells to fight diseases like cancer. Further analysis of the link between Myc and amino acid transporters could in the future aid the design of such immunotherapies.

whether or not Myc acts a general amplifier of active gene transcription (*Lewis et al., 2018*; *Lin et al., 2012*; *Nie et al., 2012*) or has more selective actions (*Sabò et al., 2014*; *Tesi et al., 2019*). There is also evidence Myc can act post transcriptionally, controlling mRNA cap methylation and broadly enhancing mRNA translation (*Cowling and Cole, 2007*; *Ruggero, 2009*; *Singh et al., 2019*). The salient point is that there appear to be no universal models of Myc action that can be applied to all cell lineages. As an example, it is reported that oncogenic Myc mutants control amino acid transporter expression in tumour cells (*Yue et al., 2017*) whereas analysis of endogenous Myc function in immune activated primary B cells found no such role (*Tesi et al., 2019*). These discrepancies highlight the neccessity for direct experimental analysis to understand how Myc controls T lymphocyte function rather than simply being able to extrapolate from other cell models. In this context, T lymphocytes are critical cells of the adaptive immune response and understanding the signalling checkpoints that control T cell function is fundamental for any strategy to manipulate T cell function for immunotherapy or immunosuppression.

T cell immune activation is associated with increases in mRNA translation, amino acid transport and protein synthesis all of which shape the execution of the T cell transcriptional program and completely reshape the T cell proteome (*Araki et al., 2017*; *Geiger et al., 2016*; *Howden et al., 2019*; *Ricciardi et al., 2018*; *Sinclair et al., 2013*). Hence, one way to gain a full and unbiased understanding of how Myc controls T cell metabolism and T cell function is an in-depth analysis of how Myc shapes T cell proteomes. Accordingly, we have used high-resolution mass-spectrometry to perform a quantitative analysis of the impact of Myc deficiency on the proteomes of immune activated CD4$^+$ and CD8$^+$ T cells. These data reveal a selectivity of Myc action in co-ordinating T cell proteomes and identify both Myc dependent and Myc independent remodelling of T cell metabolic

programs. The data uncover that a primary function of Myc in T lymphocytes is control of amino acid transporter expression which affords new insight about how Myc controls T cell biosynthetic and bio-energetic programs.

## Results

### Selective remodelling of T cell proteomes by Myc

To explore how Myc controls T cell function we used a Cd4Cre⁺Myc^{fl/fl} (Myc^{cKO}) mouse model in which *Myc* is conditionally deleted during late thymic development (*Dose et al., 2009*; *Mycko et al., 2009*; *Trumpp et al., 2001*). As shown previously (*Wang et al., 2011*), Myc-deficient CD4⁺ and CD8⁺ T cells do not substantially increase cell size or proliferate in response to immune activation with anti-CD3/anti-CD28 agonist antibodies (*Figure 1A*, *Figure 1—figure supplement 1A*). To examine how Myc loss impacts proteome remodelling during immune activation we per-formed quantitative label-free high-resolution mass spectrometry on 24 hour CD3/CD28 activated wild-type (Cd4Cre ⁺, Myc^{WT}) and Myc^{cKO} CD4⁺ and CD8⁺ T cells. This time point was chosen as it is when we observe maximal increase in cell size of the immune activated cells with no difference in survival between Myc^{WT} and Myc^{cKO} T cells. Moreover, at this time point there is minimal impact of autocrine secreted cytokine IL-2 on Myc expression (*Figure 1—figure supplement 1B*).

>7000 proteins were identified and protein mass and copy number per cell was estimated by the 'proteomic ruler' method which uses the mass spectrometry signal of histones as an internal stan-dard (*Supplementary file 1*; *Wiśniewski et al., 2014*). The data in *Figure 1B* show that in contrast to Myc T cells, CD3/CD28 activated Myc^{cKO} T cells fail to increase protein content above the level of naive ex vivo isolated Myc^{WT} T cells. Hence, the increase in cell biomass that accompanies T cell acti-vation is dependent on Myc. Notably, the protein content of immune activated Myc^{WT} CD4⁺ T cells was lower than activated CD8⁺ T cells and this correlates with higher levels of Myc in immune acti-vated CD8⁺ versus CD4⁺ Myc^{WT} T cells (*Figure 1C–D*).

The immune activation of T cells is accompanied by complex proteome remodelling (*Geiger et al., 2016*; *Howden et al., 2019*; *Ron-Harel et al., 2016*; *Tan et al., 2017*). A key ques-tion is whether the dramatically lower cell mass in CD3/CD28 activated Myc^{cKO} T cells reflects a scaled decrease in expression of all proteins or a selective loss of protein expression. In this respect a few hundred very abundant proteins are known to account for most cellular mass (*Howden et al., 2019*; *Hukelmann et al., 2016*; *Ly et al., 2014*), with 75% of the protein mass of immune activated Myc^{WT} CD4⁺ and CD8⁺ T cells comprising 344 and 391 proteins respectively. Myc-deficiency reduced the expression of most, but not all of these abundant proteins (*Figure 1E–F*). To assess the selectivity of Myc control of T cell proteomes we used nearest neighbour analysis and Pearson corre-lation to group and align the expression profile of ~6400 proteins in naïve and immune activated Myc^{WT} and Myc^{cKO} CD4⁺ and CD8⁺ T cells (*Figure 1G*). These analyses highlight how CD3/CD28 stimulation dynamically reshapes the proteomic landscape of CD4⁺ and CD8⁺ T cells. The impact of Myc loss is striking but clearly selective and not a simple scaled decrease in expression of all pro-teins. There are a number of proteins expressed at high levels in naïve cells and downregulated by immune activation in both Myc^{WT} and Myc^{cKO} T cells (*Figure 1G*), including Kruppel family transcrip-tion factors which maintain pluripotency and cell quiescence (eg Klf2) and growth factors receptors such as the IL7 receptor (*Figure 1H–I*). There is also a subset of ~300–450 proteins that are strongly induced by immune activation irrespective of Myc expression (*Figure 1G*). These include CD69, CD44 and transcription factors cRel and JunB (*Figure 1J–M*). The critical transcription factors T-bet and Irf4 were also upregulated in immune Myc^{cKO} T cells, albeit at reduced levels compared with Myc^{WT} T cells (*Figure 1N–O*).

The selective effects of Myc-deficiency on protein expression in activated CD8⁺ and CD4⁺ T cells appeared qualitatively similar (*Figure 1G*). There were however some quantitative differences. These differences reflect that some proteins were more highly expressed in activated Myc^{WT} CD8⁺ T cells than in Myc^{WT} CD4⁺ T cells, however, Myc-deficiency reduced protein expression down to a similar level in both CD4⁺ and CD8⁺ T cells, therefore giving a larger effect size in CD8⁺ T cells (*Figure 1—figure supplement 2*). When taken in conjunction with the observation that CD8⁺ T cells expressed a higher level of Myc (*Figure 1C–D*), associated with increased cell biomass (*Figure 1A–B*), this sug-gests a dose-dependent Myc-driven amplification of protein expression.

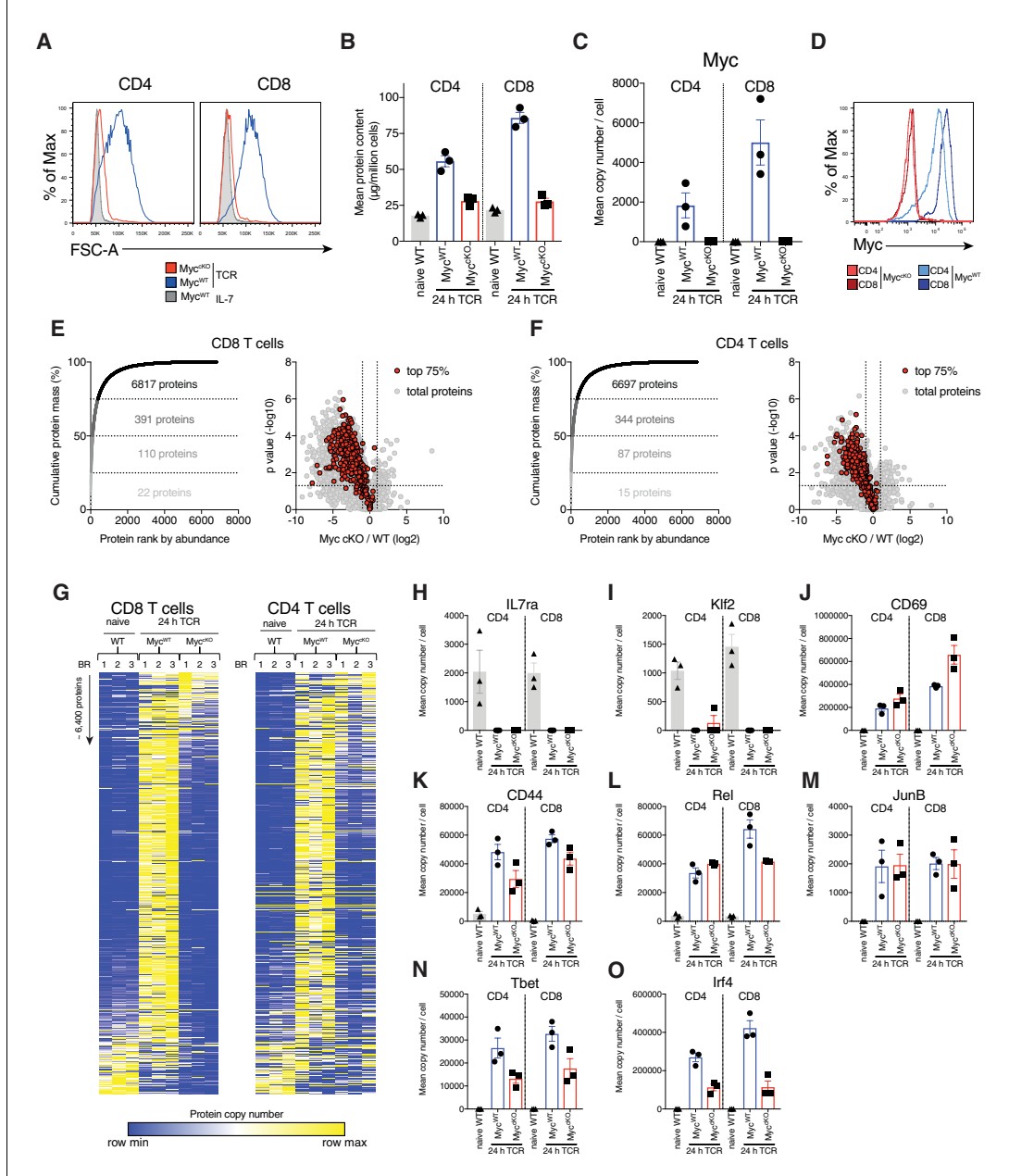

**Figure 1.** Myc controls cell growth by selectively remodelling T cell proteomes. (**A**) Forward scatter area (FSC-A) of IL-7 maintained or 24 hr anti-CD3 + anti-CD28 (TCR) activated Cd4*Cre*[+] (Myc[WT]) and Cd4*Cre*[+]*Myc*[fl/fl] (Myc[cKO]) T cells. (**B–C, E–O**) Quantitative proteomics data of ex vivo naïve WT and 24 hr TCR activated CD4[+] and CD8[+] T cells from Myc[WT] and Myc[cKO] mice. (**B**) Total protein content (μg/million cells). (**C**) Mean protein copy number per cell estimated using proteomic ruler (*Wiśniewski et al., 2014*) of Myc. (**D**) Myc expression measured by flow cytometry in 24 hr TCR activated Myc[WT] and Myc[cKO] CD4[+] and CD8[+] T cells. Proteins from 24 hr TCR activated Myc[WT] (**E**) CD8[+] and (**F**) CD4[+] T cells were ranked by mass contribution and the mean cumulative protein mass was plotted against protein rank (left panel). Numbers in each quartile indicate total proteins summed with those in the quartiles below. Volcano plots show foldchange in protein copy number between TCR activated Myc[cKO] and Myc[WT] T cells, with proteins that contribute the top 75% of the T cell mass shown in red (right panel). (**G**) Heat maps of naïve and TCR activated Myc[WT] and Myc[cKO] CD8[+] and CD4[+] proteomes. Relative protein abundance is graded from low (blue) to high (yellow) per row. Input data for heatmaps is listed in *Supplementary file 1*. Mean protein copy number per cell for activation markers (**H**) IL7ra (**J**) CD69 and (**K**) CD44 and key transcription factors (**I**) Klf2, (**L**) Rel, (**M**) JunB, (**N**) Tbet, and (**O**) Irf4. Symbols on bar charts represent biological replicates: error bars show mean ± S.E.M. Quantitative proteomics was performed on biological triplicates. Fold-change calculations and statistical testing comparing naïve WT vs TCR Myc[WT], naïve WT vs TCR Myc[cKO], and TCR Myc[WT] vs TCR Myc[cKO] protein copy number per cell is listed in *Supplementary file 1*.

The online version of this article includes the following figure supplement(s) for figure 1:

**Figure supplement 1.** Immune activated Myc-deficient T cells fail to proliferate.

*Figure 1 continued on next page*

*Figure 1 continued*

**Figure supplement 2.** Myc-deficiency has a larger quantitative effect in CD8[+] T cells.

Collectively, these data show that immune activated T cell proteome remodelling comprises both Myc dependent and independent processes and that Myc has a qualitatively similar, but dose-dependent effect on CD4[+] and CD8[+] T cell proteomes.

## Selective remodelling of T cell metabolic pathways by Myc

When examining the selective effects of Myc-deficiency on T cell immune activation we observed that Myc[cKO] T cells increased expression of the glucose transporters Slc2a1 and Slc2a3 (Glut1 and Glut3 respectively) equal to, or exceeding the level seen in Myc[WT] T cells in response to T cell activation (*Figure 2A–B*). The ability of immune activated Myc[cKO] T cells to upregulate expression of Slc2a1 and Slc2a3 glucose transporters was unexpected as it has been reported that Myc-deficient T cells have abnormal glycolytic metabolism and defective induction of glucose transporter mRNA (*Wang et al., 2011*). Moreover, *Slc2a1* has been implicated as a direct transcriptional target of Myc (*Osthus et al., 2000*). In this context, CD3/CD28 triggering increases expression of glycolytic enzymes in both Myc[WT] and Myc[cKO] CD4[+] and CD8[+] T cells (*Figure 2C*, left panel). Although the cumulative levels of glycolytic enzymes in Myc[cKO] are reduced by 58% and 30% in CD8[+] and CD4[+] T cells respectively compared with Myc[WT] controls, they still comprise a large percentage of the proteomes of immune activated Myc[cKO] T cells (*Figure 2C*, right panel). It was however striking that Myc had a large impact on lactate transporter expression, particularly on the numerically dominant lactate transporter Slc16a1 (*Figure 2D*). Lactate transporters control a critical rate limiting step for glycolytic flux (*Tanner et al., 2018*). Their absence would prevent lactate export and feedback to suppress glycolytic flux (*Doherty et al., 2014*). Slc16a1 expression increases from <10,000 copies per naïve T cell to ~140,000 and ~80,000 copies per immune activated CD8[+] and CD4[+] Myc[WT] T cell respectively. In contrast, Slc16a1 expression in immune activated Myc[cKO] T cells remains equivalent to naive levels (*Figure 2D*, *Supplementary file 1*). These data display the selectivity of Myc importance for expression of key components of the glycolysis machinery and point to control of lactate export as a mechanism whereby Myc controls glycolytic flux in T cells.

Another key Myc controlled metabolic process is glutamine catabolism (*Wang et al., 2011*; *Wise et al., 2008*). Once imported glutamine can be metabolised in a number of different processes, including the hexosamine pathway, nucleotide biosynthesis processes, and the citric acid cycle (*Figure 2E*). The present data reveal the selectivity of the Myc requirement for expression of important enzymes for glutamine metabolism. Myc controls expression of glutaminase (Gls), Cad and Ppat, the enzymes that control the first steps in glutaminolysis, and pyrimidine and purine biosynthesis respectively. However, expression of Gfpt1, the first and rate limiting step in the hexosamine pathway and Glud1, the enzyme that converts glutamate to a-ketoglutarate are still expressed in Myc[cKO] T cells (*Figure 2F* and *Supplementary file 1*).

## Myc controls amino acid transporter expression in immune activated T cells

One major effect of Myc loss on immune activated T cells is failure to increase cell mass (*Figure 1A–B*). In this context, immune activation of T cells decreases expression of translational repressors and drives increased expression of ribosomes and mRNA translational machinery (*Geiger et al., 2016*; *Howden et al., 2019*; *Ron-Harel et al., 2016*; *Tan et al., 2017*). The data in *Figure 3—figure supplement 1A–C* shows that Myc loss does not prevent loss of the translational repressor Pdcd4 in activated T cells. Myc-deficiency did however suppress CD3/CD28 mediated increases in expression of ribosomes, eukaryotic initiation factor 4 (eIF4F) complexes that translate methyl capped mRNAs and EIF2 complexes which controls tRNA transfer to ribosomes. Although increasing expression of translational machinery is important, an absolutely fundamental requirement for a substantial increase in cell mass is availability of amino acids (*Hosios et al., 2016*). Therefore, it is striking that the loss of Myc prevents the upregulation of expression of multiple amino acid transporters in activated T cells (*Figure 3A–B*). The most abundant amino acid transporters expressed on CD3/CD28 activated CD4[+] and CD8[+] T cells are Slc7a5 (leucine, methionine, tryptophan), Slc1a5 (glutamine,

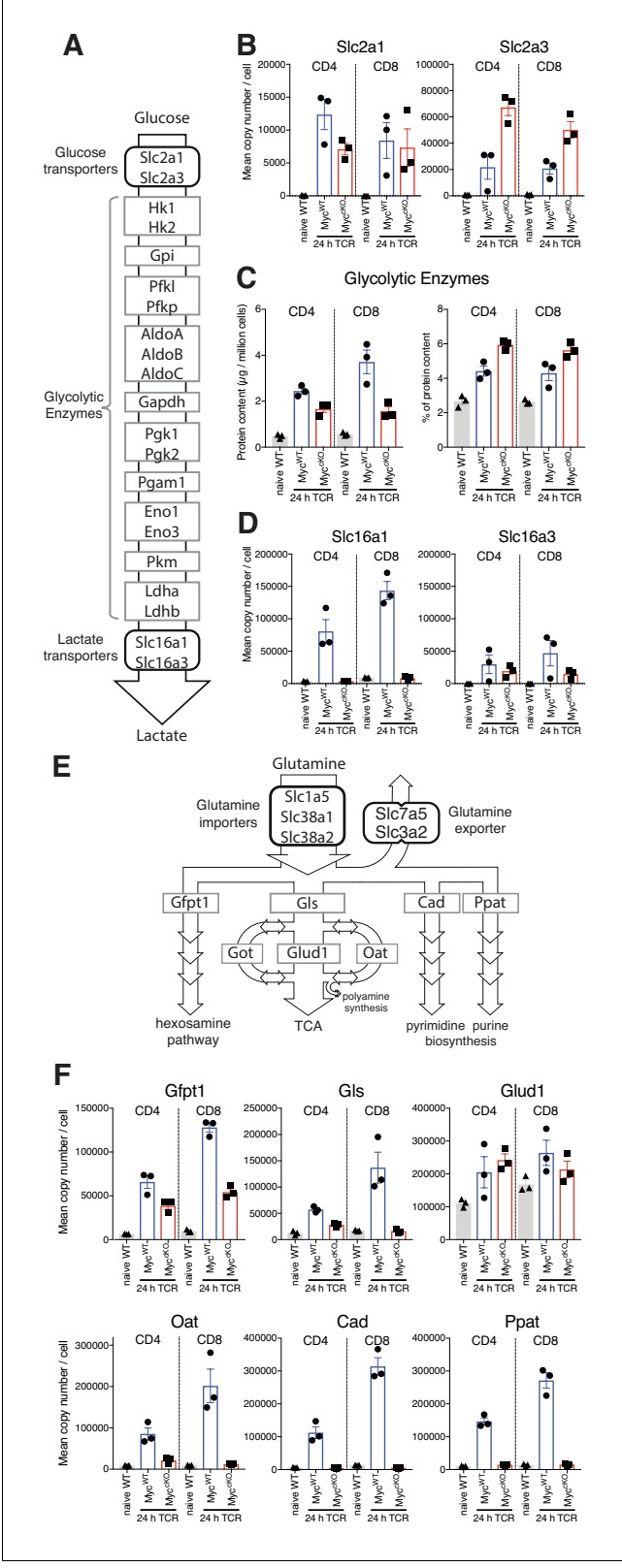

**Figure 2.** Myc control of T cell metabolism is selective. Naïve WT and 24 hr TCR activated Myc^WT and Myc^cKO CD4^+ and CD8^+ T cell proteomic data was generated as described in *Figure 1* and Materials and methods. (**A**) schematic of nutrient transporters and enzymes involved in glycolysis. (**B**) Mean copy number per cell for glucose transporters Slc2a1 and Slc2a3. (**C**) Total protein content (μg/million cells) (left panel) and % contribution to total

*Figure 2 continued on next page*

*Figure 2 continued*

cellular protein mass (right panel) of total glycolytic enzymes. (D) Mean protein copy number per cell for lactate transporters Slc16a1 and Slc16a3. (E) Schematic of transporters and enzymes involved in Glutamine transport and metabolism. (F) Mean copy number per cell for major enzymes involved in glutamine metabolism. Symbols on bar charts represent biological replicates from biological triplicate data, error bars show mean ± S.E.M. Fold-change calculations and statistical testing comparing naïve WT vs TCR Myc$^{WT}$, naïve WT vs TCR Myc$^{cKO}$, and TCR Myc$^{WT}$ vs TCR Myc$^{cKO}$ protein copy number per cell is listed in *Supplementary file 1*.

serine, threonine, alanine), Slc38a1 and Slc38a2 (glutamine, methionine) and Slc7a1(arginine, lysine) (*Figure 3A*, *Supplementary file 1*). Naïve T cells have very low levels of all of these transporters, expressing ~500–2500 copies per cell (*Figure 3A*, *Supplementary file 1*). Upon activation, amino acid transporters are some of the most highly induced proteins in Myc$^{WT}$ T cells, exhibiting up to 100-fold increases relative to naïve cells (*Figure 3A–B*). In contrast, immune activated Myc$^{cKO}$ T cells only express amino acid transporters at near naïve levels (*Figure 3A–B*, *Supplementary file 1*).

The high levels of protein production in activated T cells would need to be fuelled by amino acid supply (*Hosios et al., 2016*). Moreover, T cells that lack expression of key amino acid transporters such as Slc7a5 and Slc1a5 are defective in their response to T cell activation (*Nakaya et al., 2014*; *Sinclair et al., 2013*). We therefore questioned whether the ability of Myc to control T cell growth could be explained by Myc control of amino acid transporter expression. Accordingly, we examined the impact of Myc expression on the functional capacity of T cells to transport amino acids and we assessed whether the loss of amino acid transporter expression could recapitulate the striking impact of *Myc* deletion on T cell protein production. We focused on the system L transporter Slc7a5, as this is the most abundant amino acid transporter expressed on immune activated T lymphocytes (*Figures 3A* and *4B*, *Supplementary file 1*) and mediates transport of many essential amino acids including methionine, leucine, isoleucine, valine, phenylalanine and tryptophan (*Sinclair et al., 2019*; *Sinclair et al., 2018*; *Sinclair et al., 2013*). Low basal levels of Slc7a5 in naïve T cells mediate amino acid uptake that is not dependent on Myc (*Figure 3C*). Within 4 hr of T cell activation there is already increased system L transport activity in Myc$^{WT}$ T cells and this increase is substantially lower in Myc$^{cKO}$ CD4$^+$ and CD8$^+$ T cells (*Figure 3D–E*). There was also a strong correlation between the levels of Myc protein expressed by activated T cells and system L amino acid transport capacity (*Figure 3F*) and while system L transport increased substantially over the first 24 hr of T cell activation in Myc$^{WT}$ T cells this did not occur in Myc$^{cKO}$ T cells (*Figure 3—figure supplement 2*). Downstream of Slc7a5 amino acid uptake, Myc-deficient T cells also fail to increase expression of several key enzymes in metabolic pathways that utilise branch-chain amino acid (Leucine, Isoleucine, Valine) pathways and methionine (*Figure 3—figure supplement 3A–B*). Collectively, these data show that Myc plays a critical role in regulating system L amino acid transport and amino acid metabolism in immune activated T cells.

Could loss of amino acid transport be the mechanism for the loss of protein production in immune activated Myc$^{cKO}$ T cells? To assess this, we examined the impact of *Slc7a5* deletion on immune activated T cell proteomes. *Figure 3G–H* shows that the dramatic increase in cell mass associated with normal T cell activation does not occur in immune activated Slc7a5$^{cKO}$ (Cd4*Cre*$^+$ *Slc7a5*$^{fl/fl}$) CD4$^+$ T cells. We then used nearest neighbour analysis and Pearson correlation to group the expression profile of ~6800 proteins from naïve wild-type and immune activated Slc7a5$^{WT}$ and Slc7a5$^{cKO}$ CD4$^+$ T cell proteomes. These data show Slc7a5 deficiency, similar to Myc deficiency, has a profound effect on protein expression in immune activated CD4$^+$ T cells (*Figure 3I*). Slc7a5$^{cKO}$ T cells still respond to antigen receptor activation to downregulate a subset of naïve T cell proteins and can still upregulate expression of a small subset of proteins (*Figure 3I*). The data show a striking overlap in proteins that were both Myc and Slc7a5 regulated (*Figure 3J*). Most of this overlap was in proteins that were reduced in response to Myc or Slc7a5 deficiency (*Figure 3K*), including translational machinery such as ribosomes (*Figure 3—figure supplement 4A–B*). Although there is a large degree of overlap in the proteomics data, Slc7a5-deficiency does not completely phenocopy the effects of Myc-deficiency. Induction of proteins such as the glucose transporter Slc2a3 (*Figure 2B*, *Figure 3—figure supplement 4C*) and effector molecules like Granzyme B and IFNγ (*Figure 3—figure supplement 4D–E*) exhibit a more severe defect in Slc7a5$^{cKO}$ T cells. This is likely due to the lack

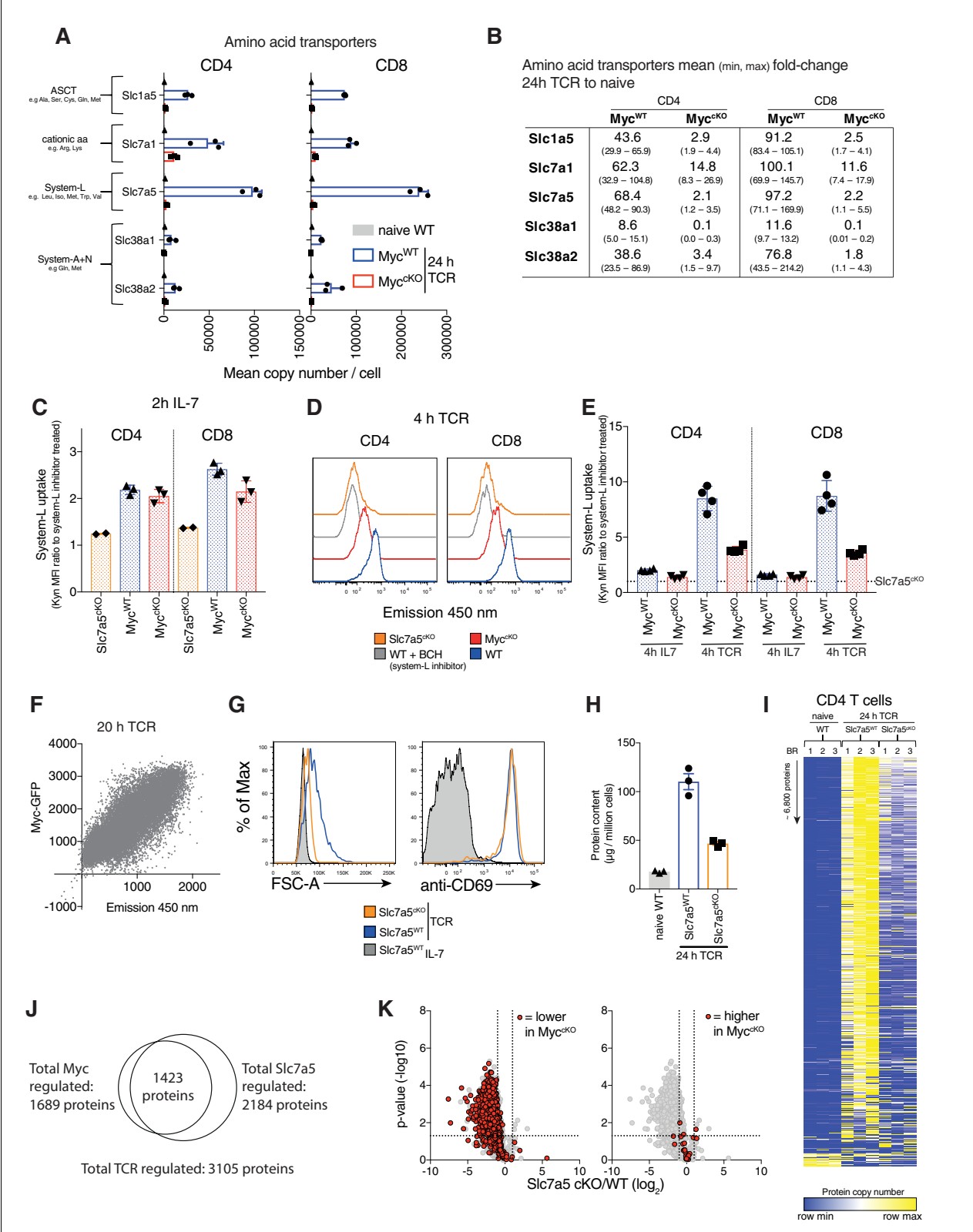

**Figure 3.** Myc induces amino acid transporter expression, a critical step for proteome remodelling. Naïve WT and 24 hr TCR activated Myc$^{WT}$ and Myc$^{cKO}$ CD4$^+$ and CD8$^+$ T cell proteomic data was generated as described in *Figure 1* and Materials and methods. (**A**) Mean copy number per cell of abundant amino acid transporters in T cells. (**B**) Fold-change in amino acid transporter protein copy number from naïve WT to 24 hr TCR activated Myc$^{WT}$ and Myc$^{cKO}$, mean (min, max). Transport by system L amino acid transporters was measured by uptake of fluorescent (emission 450 nm when

*Figure 3 continued on next page*

*Figure 3 continued*

excited at 405 nm) Tryptophan metabolite, Kynurenine (Kyn) (*Sinclair et al., 2018*) in (C) 2 hr IL-7 maintained and (D-E) 4 hr IL-7 maintained or TCR activated splenic CD4$^+$ and CD8$^+$ WT, Myc$^{cKO}$ and Slc7a5$^{cKO}$ T cells or (F) 20 hr TCR activated Myc-GFP reporter CD4$^+$ T cells. In (C,E) system-L uptake is represented as the ratio of BCH (a system L inhibitor) untreated: treated T cells. In (E) dotted line indicates Slc7a5$^{cKO}$ uptake level. (G) Forward scatter and CD69 expression of IL-7 maintained or 24 hr TCR activated wild-type and Slc7a5$^{cKO}$ (Cd4*Cre*$^+$ *Slc7a5*$^{fl/fl}$) T cells. (H-K) Quantitative proteomics data of naïve WT and 24 hr TCR activated CD4$^+$ and CD8$^+$ T cells from Ly5.1 (Slc7a5$^{WT}$) and Slc7a5$^{cKO}$ mice. Baseline naïve WT data is the same as used for the Myc$^{cKO}$ dataset. (H) Total protein content (µg/million cells). (I) Heat map of naïve and TCR activated Slc7a5$^{WT}$ and Slc7a5$^{cKO}$ CD4$^+$ T cell proteomes. Relative protein abundance is graded from low (blue) to high (yellow) per row. Input data for heatmaps is listed in *Supplementary file 1*. (J) Venn diagram showing the overlap in TCR regulated proteins that are more than 2-fold regulated and p<0.05 in Myc$^{WT}$ vs Myc$^{cKO}$ and Slc7a5$^{WT}$ vs Slc7a5$^{cKO}$ TCR activated CD4$^+$ T cells. (K) Volcano plots of TCR regulated proteins comparing Slc7a5$^{WT}$ and Slc7a5$^{cKO}$ datatsets. Proteins > 2 fold different between Myc$^{WT}$ and Myc$^{cKO}$ TCR activated T cells are highlighted in red; proteins reduced in the Myc$^{cKO}$ (left panel), proteins higher Myc$^{cKO}$ (right panel). Symbols in bar charts represent biological replicates: error bars show mean ± S.E.M. Dot plot in (F) is representative of biological triplicate data. Quantitative proteomics was performed on biological triplicates. Fold-change calculations and statistical testing comparing naïve WT vs TCR Myc$^{WT}$, naïve WT vs TCR Myc$^{cKO}$, TCR Myc$^{WT}$ vs TCR Myc$^{cKO}$, naïve WT vs TCR Slc7a5$^{WT}$, naïve WT vs TCR Slc7a5$^{cKO}$ and TCR Slc7a5$^{WT}$ vs TCR Slc7a5$^{cKO}$ protein copy number per cell is listed in *Supplementary file 1*.

The online version of this article includes the following figure supplement(s) for figure 3:

**Figure supplement 1.** Myc-deficient T cells fail to induce protein translation machinery.
**Figure supplement 2.** Amino acid transport capacity corresponds with transporter number.
**Figure supplement 3.** Myc-deficient T cells fail to induce Branched-chain amino acid and Methionine metabolism.
**Figure supplement 4.** Ribosome expression is reduced in both Myc and Slc7a5 deficient T cells, but other proteins are differentially regulated between Myc and Slc7a5 deficient T cells.

of basal-level amino acid transport in naïve Slc7a5$^{cKO}$ T cells which is not deficient in naïve Myc$^{cKO}$ T cells (*Figure 3C*).

Overall, deficiency in a single Myc controlled amino acid transporter, Slc7a5, largely does mimic the phenotype of Myc$^{cKO}$ T cells, preventing T cell growth and selectively controlling proteome remodelling.

To explore the mechanism for Myc control of amino acid transport in activated T cells we examined the relationship between Myc and amino acid transporter mRNA expression. Single cell RNA-seq analysis of antigen activated OT1 CD8$^+$ T cells (*Richard et al., 2018*) shows a strong correlation at the single cell level of *Myc* mRNA expression and expression of mRNA for *Slc7a5*, *Slc7a1* and *Slc1a5* (*Figure 4A*). Expression of *Myc* mRNA clearly precedes increased expression of mRNA for *Slc7a5* and *Slc1a5* (*Figure 4A*). More importantly, in a proteomics time course of OT1 CD8$^+$ T cell activation, expression of Myc protein precedes antigen induced increases in expression of most amino acid transporters (*Figure 4B*). Proteomics data shows that expression of amino acid transporters increases gradually over time (*Figure 4B*), and Kyn uptake experiments confirm that this increase in transporter number corresponds with higher system L uptake (*Figure 3—figure supplement 2*). CD3/CD28 activation of Myc$^{WT}$ CD4$^+$ and CD8$^+$ T cells drives increases in *Slc7a5, Slc1a5* and *Slc7a1* mRNA, whereas activated Myc$^{cKO}$ CD4$^+$ and CD8$^+$ T cells do not increase expression of *Slc7a5* or *Slc1a5* mRNA and show reduced expression of *Slc7a1* mRNA (*Figure 4C*).

## Myc induction of amino acid transport initiates a positive feedforward loop

The current data are consistent with a model that Myc controls T cell growth by controlling the upregulation of amino acid transporter expression required for T cell activation. However one possible inconsistency is that previous studies have shown that Slc7a5 is required for expression of Myc protein (but not mRNA) in activated CD8$^+$ T cells (*Sinclair et al., 2013*). We considered that an explanation for this discrepancy would be if there were a positive feedforward loop whereby the initial rapid expression of Myc during immune activation is not Slc7a5 dependent but the sustained expression is. To directly interrogate this model we measured Myc expression over time in CD3/CD28 activated WT and Slc7a5$^{cKO}$ T cells. These data (*Figure 4D*) show that Slc7a5 is not required for the immediate and rapid upregulation of Myc expression that accompanies T cell activation but is required for activated T cells to sustain Myc protein.

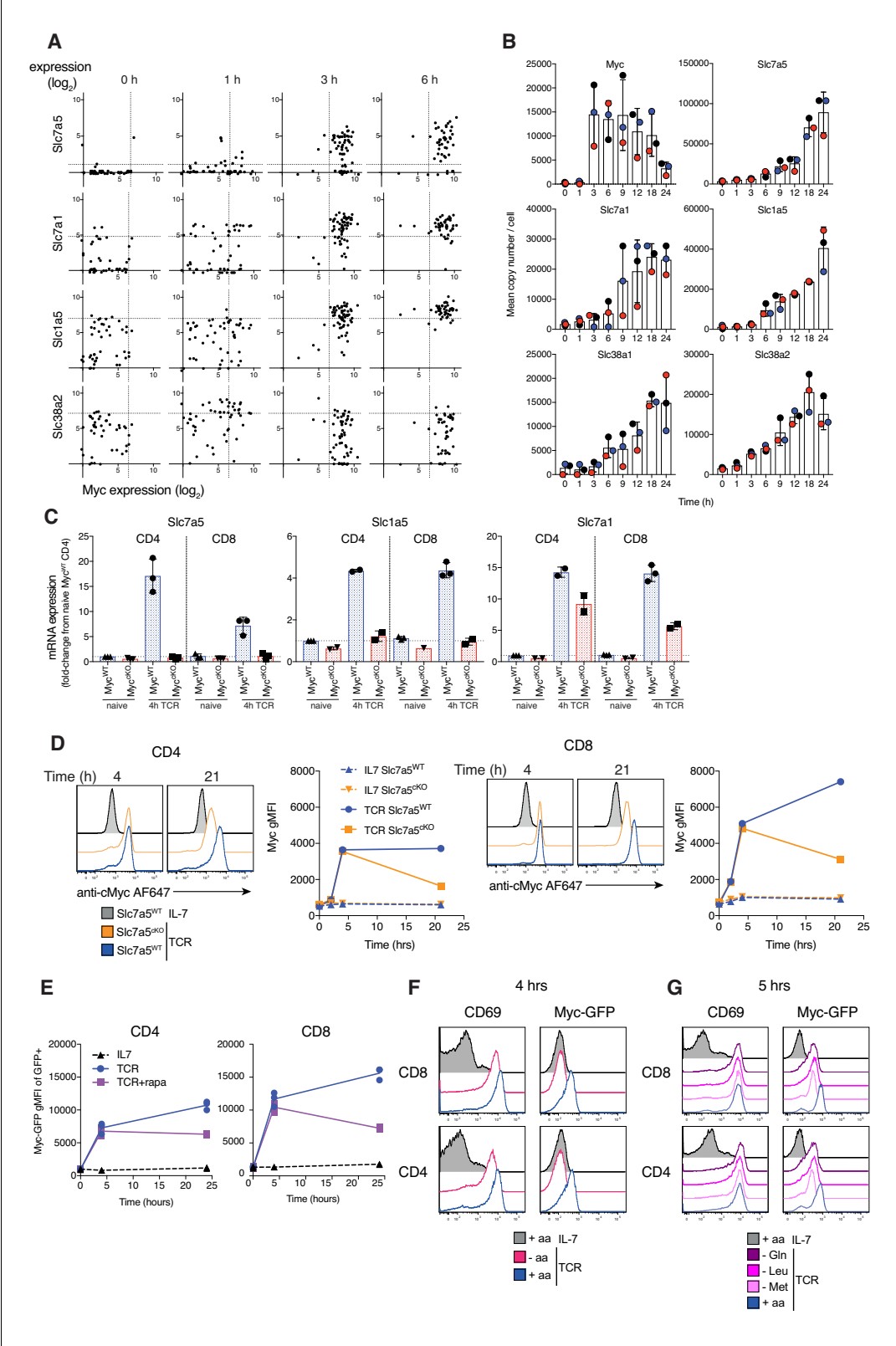

**Figure 4.** Myc induces amino acid transport early after TCR activation, triggering a feedforward loop maintaining its own expression. (**A**) Expression levels of *Slc7a5, Slc7a1, Slc1a5* and *Slc38a2* vs *Myc* mRNA from published single cell RNAseq dataset of OT1 T cells stimulated with SIINFEKL (N4) peptide for the indicated times (***Richard et al., 2018***). Dotted lines represent 95th percentile of 0 hr gene expression. Symbols represent individual cells (**B**) Quantitative proteomics of OT-I CD8[+] T cells activated with N4 peptide for the indicated time. Mean copy number per cell of the proteins Myc,

*Figure 4 continued on next page*

*Figure 4 continued*

Slc7a5, Slc7a1, Slc1a5, Slc38a1 and Slc38a2. (C) *Slc7a5, Slc1a5* and *Slc7a1* mRNA measured by qPCR from ex vivo naïve or 4 hr TCR activated lymph node CD4+ and CD8+ T cells. mRNA levels are relative to naïve CD4+ T cells. (D) Histograms and geometric mean fluorescence intensity (gMFI) vs time of Myc protein measured with antibody by flow cytometry in IL-7 maintained or TCR activated Slc7a5WT and Slc7a5cKO lymph node CD4+ and CD8+ T cells. Representative of 4 biological replicates. (E) Myc-GFP gMFI in IL-7 maintained or GFP+ TCR activated CD4+ and CD8+ T cells ± rapamycin from lymph nodes of Myc-GFP reporter mouse. Myc-GFP reporter expression in CD4+ and CD8+ T cells maintained in IL-7 or TCR activated in (F) amino-acid free media (-aa, HBSS), vs RPMI (+aa) or (G) media deficient in a single amino acid. Data representative of at least three biological replicates. Symbols unless otherwise stated represent biological replicates. Mean ± S.E.M. Quantitative proteomics was performed on biological triplicates.

The finding that Slc7a5cKO T cells can induce Myc expression is also surprising in the context of previous work demonstrating an important role for mTORC1 activation (which is critically dependent on uptake of leucine) in controlling Myc expression during T cell activation (*Wang et al., 2011*). Therefore, we tested the dependency of Myc protein expression on mTORC1 signalling in our T cell system. The data show that rapamycin treatment had very little impact on expression of Myc protein after 4 hr of T cell activation but did reduce Myc expression after 24 hr (*Figure 4E*), consistent with the results seen in Slc7a5cKO T cells (*Figure 4D*). Although Slc7a5 deficiency or mTORC1 inhibition alone was insufficient to prevent Myc induction, the presence of external amino acids is necessary for expression of Myc protein, with T cells activated in amino acid-free media being unable to express Myc protein despite increasing the activation marker CD69 (*Figure 4F*). Deficiency of a single amino acid from the media, such as glutamine, methionine or leucine leads to a reduction in Myc levels but does not prevent its expression (*Figure 4G*).

Myc driven increase in amino acid transport thus triggers a positive feedforward loop supporting its own continued expression which in turn drives and sustains further increases in amino acid transport.

## Discussion

This study has mapped the impact of Myc deletion on antigen driven proteome remodelling of CD4+ and CD8+ T cells to understand how Myc controls T cell activation and metabolic reprogramming. The study uncovers both Myc dependent and independent restructuring of the T cell proteome during immune activation. Myc was required for the increase in expression of important metabolic pathway proteins; for example, glutamine transporters and glutaminase, key proteins controlling the first steps of glutaminolysis (*Newsholme et al., 1985*); and lactate transporters, a major rate determining step for glycolytic flux (*Tanner et al., 2018*). However, the current data also show that expression of many key metabolic enzymes for both glutaminolysis and glycolysis can still occur in immune activated Myc null T cells. In particular, in the context of glucose metabolism an unexpected observation was that Myc was not required for protein expression of the glucose transporters Slc2a1 and Slc2a3 in activated T cells. This was surprising, given previous observations that Myc deletion reduced *Slc2a1* and *Slc2a3* mRNA (*Wang et al., 2011*) and highlights the value of a proteomics approach to quantify the expression patterns of proteins where there may be a disconnect between mRNA and protein expression due to translational regulation (*Ricciardi et al., 2018*). The expression of both glucose and lactate transporters are key for glycolytic flux (*Tanner et al., 2018*) and the fact that these are differentially controlled by Myc reveals that upregulation of metabolic pathways during T cell activation is more complex than a simple activation switch or amplifier mediated by a single transcription factor (*Nie et al., 2012*). The data gives molecular insight into why Myc is so important for T cell glutamine metabolism and glycolysis but they also reveal that T cell metabolic reprogramming requires the coordination of Myc expression with other signalling pathways. In this respect we have shown recently that activation of mTORc1 is not required for Myc expression in activated T cell but does have a substantive effect on the expression of glucose transporter protein (*Howden et al., 2019*).

One key conclusion from the present data is that a primary function of Myc is to control expression of the amino acid transporters, inducing a positive feedforward loop to sustain Myc levels in activated T cells. A salient point is that Myc was only necessary for immune activation associated increases in amino acid transporter expression. The absence of Myc did not impinge on the low

basal levels of amino acid transport through the system L transporter seen in naïve T cells and it was clear from the proteomic data that Myc null T cells still had some capacity to increase expression of key proteins. The inability of Myc null T cells to increase amino acid transport aligns with previous metabolomic data that Myc null cells have decreased levels of intracellular amino acids (*Wang et al., 2011*). The loss of amino acid transporter induction in Myc null T cells would also prevent the increases in expression of the protein biosynthetic machinery as well as preventing uptake of the raw material required to synthesise protein. The importance of Myc induction of amino acid transporters for T cell activation is particularly highlighted by the large effect of deleting just one of the Myc controlled amino acid transporters, Slc7a5, on the T cell proteome, which almost phenocopies the effects of Myc deletion itself. The impact of the loss of a single Myc controlled amino acid transporter was remarkable and reflects that Slc7a5 transports multiple large neutral amino acids including Leucine, Phenylalanine and Tryptophan. Myc control of Slc7a5 expression would be particularly important for protein synthesis as Slc7a5 is also the major T cell transporter for Methionine, the predominant 'start' amino acid used to initiate polypeptide synthesis during mRNA translation (*Sinclair et al., 2019*; *Sinclair et al., 2013*). These data highlight how Myc control of even one amino acid transporter, Slc7a5, would have indirect consequences for the expression of thousands of proteins in immune activated T cells and could underpin the ability of Myc to regulate multiple biosynthetic, bioenergetic and epigenetic processes in T cells.

# Materials and methods

## Key resources table

| Reagent type (species) or resource | Designation | Source or reference | Identifiers | Additional information |
|---|---|---|---|---|
| Genetic reagent (*M. musculus*) | Cd4*Cre* | PMID: 27345256, PMID: 11728338 | | |
| Genetic reagent (*M. musculus*) | Cd4*Cre*[+] *Myc*[fl/fl] | PMID: 19423665, PMID: 19342639, PMID: 11742404 | | |
| Genetic reagent (*M. musculus*) | Cd4*Cre*[+] *Slc7a5*[fl/fl] | PMID: 23525088, PMID: 24586861 | | |
| Genetic reagent (*M. musculus*) | GFP-Myc[KI] | PMID: 18196519, PMID: 26136212, PMID: 23021216 | | |
| Genetic reagent (*M. musculus*) | OT1 | PMID: 8287475 | | maintained in house as an OT1 TCR transgene heterozygote on a CD45.1 (Ly5.1) background |
| Antibody | Anti-CD3 (armenian hamster, monoclonal) | Thermo Fisher Scientific | Cat # 14-0031-82, RRID:AB_467049 | T cell activation: 0.5 or 1 µg/ml as indicated in Materials and methods |
| Antibody | Anti-CD28 (syrian hamster, monoclonal) | Thermo Fisher Scientific | Cat # 16-0281-82, RRID:AB_468921 | T cell activation: 0.5 or 3 µg/ml as indicated in Materials and methods |
| Antibody | Anti-CD4 (rat, monoclonal) | BD Biosciences | Cat # 553650, RRID:AB_394970; Cat# 552775, RRID:AB_394461; Cat# 553047, RRID:AB_394583 | cell surface staining 1:200 |
| Antibody | Anti-CD4 (rat, monoclonal) | Thermo Fisher Scientific | Cat# 47-0042-82, RRID:AB_1272183 | cell surface staining 1:200 |
| Antibody | Anti-CD8a (rat, monoclonal) | Biolegend | Cat# 100708, RRID:AB_312747; Cat# 100722, RRID:AB_312761 Cat# 100738, RRID:AB_11204079 | cell surface staining 1:200 |

*Continued on next page*

*Continued*

| Reagent type (species) or resource | Designation | Source or reference | Identifiers | Additional information |
|---|---|---|---|---|
| Antibody | Anti-CD8a (rat, monoclonal) | BD Biosciences | Cat# 551162, RRID:AB_394081 | cell surface staining 1:200 |
| Antibody | Anti-CD69 (armenian hamster, monoclonal) | ThermoFisher Scientific | Cat# 17-0691-82, RRID:AB_1210795 | cell surface staining 1:200 |
| Antibody | Anti-CD69 (armenian hamster, monoclonal) | Biolegend | Cat# 104514, RRID:AB_492843 | cell surface staining 1:200 |
| Antibody | Anti-CD69 (armenian hamster, monoclonal) | BD Biosciences | Cat# 553237, RRID:AB_394726 | cell surface staining 1:200 |
| Antibody | Anti-B220 (rat, monoclonal) | BD Biosciences | Cat# 553087, RRID:AB_394617 | cell surface staining 1:200 |
| Antibody | Anti-NK1.1 (mouse, monoclonal) | Biolegend | Cat# 108706, RRID:AB_313393 | cell surface staining 1:200 |
| Antibody | Anti-CD11b (rat, monoclonal) | Biolegend | Cat# 101206, RRID:AB_312789 | cell surface staining 1:200 |
| Antibody | Anti-CD25 (rat, monoclonal) | BD Biosciences | Cat# 553072, RRID:AB_394604 | cell surface staining 1:200 |
| Antibody | Anti-CD62L (rat, monoclonal) | Thermo Fisher Scientific | Cat# 12-0621-83, RRID:AB_465722 | cell surface staining 1:200 |
| Antibody | Anti-TCRb (armenian hamster, monoclonal) | Thermo Fisher Scientific | Cat# 45-5961-82, RRID:AB_925763 | cell surface staining 1:200 |
| Antibody | Anti-CD44 (rat, monoclonal) | BD Biosciences | Cat# 559250, RRID:AB_398661 | cell surface staining 1:200 |
| Antibody | Anti-Thy1.2 (rat, monoclonal) | BD Biosciences | Cat# 553006, RRID:AB_394545 | cell surface staining 1:200 |
| Antibody | Anti-CD45.1 (mouse, monoclonal) | Biolegend | Cat # 110714, RRID:AB_313503 | cell surface staining 1:200 |
| Antibody | Anti-CD45.2 (mouse, monoclonal) | Biolegend | Cat # 109816, RRID:AB_492868 | cell surface staining 1:200 |
| Antibody | Anti-mouse CD16/CD32 Fc Block, (rat, monoclonal) | BD Biosciences | Cat # 553141, RRID:AB_394656 | Fc block 1:100 |
| Antibody | c-Myc (D84C12) XP (rabbit, monoclonal) | Cell Signaling Technologies | Cat# 5605, RRID:AB_1903938 | intracellular staining 1:200 |
| Antibody | Anti-rabbit A647 (goat) | Cell Signaling Technologies | Cat # 4414, RRID:AB_10693544 | intracellular staining 1:1000 |
| Antibody | Anti-IFNg (rat, monoclonal) | Biolegend | Cat # 505810, RRID:AB_315404 | intracellular cytokine staining 1:100 |
| Antibody | Anti-Granzyme B | Thermo Fisher Scientific | Cat# 17-8898-82, RRID:AB_2688068 | intracellular cytokine staining 1:200 |
| Chemical compound, drug | DAPI | Thermo Fisher Scientific | D1306 | 1 µg/mL |
| Chemical compound, drug | Kynurenine | Sigma | Cat# K8625 | 200 µM |
| Chemical compound, drug | BCH | Sigma | Cat# A7902 | 10 mM |
| Chemical compound, drug | Rapamycin | Merck/Calbiochem | Cat# 553211 | 20 nM |
| Commercial assay or kit | Rneasy minikit | Qiagen | Cat # 74104 | |
| Commercial assay or kit | iScript cDNA Synthesis kit | Biorad | Cat#1708891 | |
| Commercial assay or kit | iTaq Universal SYBRGreen Supermix | Biorad | Cat# 1725121 | |

*Continued on next page*

*Continued*

| Reagent type (species) or resource | Designation | Source or reference | Identifiers | Additional information |
|---|---|---|---|---|
| Commercial assay or kit | EZQ protein quantitation kit | Thermo Fisher Scientific | R33200 | |
| Commercial assay or kit | Sera-Mag SpeedBead Carboxylate-modified magnetic particles (hydrophilic) | GE Lifesciences | cat# 45152105050250 | |
| Commercial assay or kit | Sera-Mag SpeedBead Carboxylate-modified magnetic particles (hydrophobic) | GE Lifesciences | cat# 65152105050250 | |
| Commercial assay or kit | CBQCA protein quantitation kit | Thermo Fisher Scientific | C6667 | |
| Commercial assay or kit | HiPPR Detergent Removal Spin Column Kit | Thermo Fisher Scientific | Cat# 88305 | |
| Commercial assay or kit | EasySep CD8 T cell isolation kit | STEMCELL Technologies, UK | Cat # 19853 | |
| Commercial assay or kit | Golgi Plug | BD Biosciences | Cat# 555029 | |
| Commercial assay or kit | eBioscience Intracellular Fixation and Permeabilization Buffer Set | Thermo Fisher Scientific | Cat# 88-8824-00 | |
| Commercial assay or kit | CFSE | Thermo Fisher Scientific/Invitrogen | Cat# C34554 | 5 µM |
| Peptide, recombinant protein | IL7 | Peprotech | Cat# 217–17 | 5 ng/mL |
| Peptide, recombinant protein | IL2 | Novartis, UK | Proleukin | 20 ng/ml |
| Peptide, recombinant protein | IL12 | Peprotech | Cat#210–12 | 2 ng/ml |
| Sequence based reagent | *Slc7a5* forward primer | | | AAG GCT GCG ACC CGT GTG |
| Sequence based reagent | *Slc7a5* reverse primer | | | ATC ACC TTG TCC CAT GTC CTT CC |
| Sequence based reagent | *Slc7a1* forward primer | | | GGA GCT TTG GC CTT CAT CAC T |
| Sequence based reagent | *Slc7a1* reverse primer | | | CAG CAC CCC AGG AGC ATT CA |
| Sequence based reagent | *Slc1a5* forward primer | | | GCC ATC ACC TCC ATC AAC GAC T |
| Sequence based reagent | *Slc1a5* reverse primer | | | AGA GCG GAA GGC AGC AGA CAC |
| Sequence based reagent | *TBP* forward primer | | | GTG AAT CTT GGC TGT AAA CTT GAC CT |
| Sequence based reagent | *TBP* reverse primer | | | CGC AGT TGT CCG TGG CTC T |
| Software, algorithm | FlowJo software | Treestar | | versions 9 and 10 |
| Software, algorithm | Maxquant | https://www.maxquant.org, PMID: 19029910 | | version 1.6.2.6 |
| Software, algorithm | Perseus | https://www.maxquant.org/perseus, PMID: 27348712 | | version 1.6.6.0 |
| Other | RPMI 1640 | Thermo Fisher Scientific/GIBCO | Cat# 21875–034 | |

*Continued*

| Reagent type (species) or resource | Designation | Source or reference | Identifiers | Additional information |
|---|---|---|---|---|
| Other | RPMI - glutamine | Thermo Fisher Scientific/GIBCO | Cat# 42401–018 | |
| Other | RPMI - methionine | DC Biosciences Ltd | | custom made RPMI without methionine and arginine - supplemented back the arginine (0.2 g/L) to RPMI levels |
| Other | RPMI - leucine | Sigma | Cat# R1780 SAFC | supplemented back arginine (0.2 g/L) and lysine (0.04 g/L)to RPMI levels |
| Other | HBSS | Thermo Fisher Scientific/GIBCO | Cat# 14025–050 | used this as amino acid-free media |
| Other | FBS | Thermo Fisher Scientific/GIBCO | Cat # 10270106 | |
| Other | FBS, dialyzed | Thermo Fisher Scientific/GIBCO | Cat# 26400044 | |
| Other | Arginine | Sigma | Cat# A5006 | |
| Other | Lysine | Sigma | Cat# L5501 | |

## Lead contact and materials availability

Further information and requests for resources and reagents should be directed to and will be fulfilled by the Lead Contact, Doreen Cantrell (d.a.cantrell@dundee.ac.uk).

## Experimental model details

### Mice

Cd4*Cre*[+], Cd4*Cre*[+] *Myc*[fl/fl] (*Dose et al., 2009*; *Mycko et al., 2009*; *Trumpp et al., 2001*), Cd4*Cre*[+] *Slc7a5*[fl/fl] (*Sinclair et al., 2013*), Ly5.1, OT1 and Myc-eGFP (*Nie et al., 2012*) mice were bred and maintained in the WTB/RUTG, University of Dundee in compliance with UK Home Office Animals (Scientific Procedures) Act 1986 guidelines. Male Cd4*Cre*[+] and Cd4*Cre*[+] *Myc*[fl/fl] and female Ly5.1 and Cd4*Cre*[+] *Slc7a5*[fl/fl] mice were used for proteomics studies at 12 weeks of age. 1 group of female and 2 groups of male OT1 mice aged 7–28 weeks were used for OT1 time course proteomics. Age/sex matched male and female mice were used for other experiments between 8–30 weeks of age.

## Method details

### Cell culture

All cells were activated and cultured at 37 ˚C with 5% $CO_2$ in complete culture medium - RPMI 1640 containing glutamine (Invitrogen), supplemented with 10% FBS (Gibco), penicillin/streptomycin (Gibco) and 50 µM β-mercaptoethanol (Sigma) unless otherwise indicated.

Single cell suspensions were generated by mashing mouse lymph nodes (brachial, axial, inguinal, superficial cervical, deep cervical, lumbar) or spleens through 70 µm strainer. Red blood cells in splenocyte suspension were lysed with 150 mM $NH_4Cl$ 10 mM $KHCO_3$110 µM $Na_2EDTA$ pH 7.8.

For 24 hr TCR activated Myc[WT] and Myc[cKO] proteomics, lymph node suspension from 2x mice per biological replicate were activated with 0.5 µg/mL anti-mouse CD3 (Biolegend) and 0.5 µg/mL anti-mouse CD28 (eBioscience) in 2 × 10 mL complete culture medium in six well plates. Samples were generated in biological triplicate.

For Slc7a5[WT] and Slc7a5[cKO] CD4[+] T cell proteomics, three biological replicates were generated. For each biological replicate, one Slc7a5[WT] and one Slc7a5[cKO] mouse was used. Slc7a5[cKO] (CD45.2) and Slc7a5[WT] (CD45.1) lymph node cells were mixed together at a ratio of 1:1 prior to 24 hr activation with 1 µg/mL anti-mouse CD3 and 3 µg/mL anti-mouse CD28 supplemented with 20 ng/mL recombinant human IL-2 (Proleukin, Novartis) and 2 ng/mL recombinant mouse IL-12 (Peprotech) at a cell density of 2 million live cells/mL.

For OT1 TCR time course proteomics CD8[+] T cells were purified from mouse spleen and lymph node single cell suspensions using EasySep mouse CD8 T cell isolation kit (STEMCell Technologies)

as per manufacturer instructions. Verification of purity indicated that live cells were 88–94% CD8$^+$. T cells were activated with 10 ng/mL SIINFEKL peptide at a density of 4 million cells / mL in 2 mL complete culture medium in 24 well plates. Cells were centrifuged at 300 rpm for 2 min before being placed in culture. Cells were harvested for proteomics at indicated time points and were washed twice with HBSS before being snap frozen in liquid nitrogen and stored at −80°C until further processing.

For qPCR, kynurenine uptake assays and Myc staining in *Figure 4D* or measurement of Myc-GFP expression in *Figure 4E–G*, lymph node or spleen single cell suspensions were activated with 1 µg/ mL anti-mouse CD3 and 3 µg/mL anti-mouse CD28 at a cell density of 4 million live cells/mL. Cells were centrifuged at 300 rpm for 2 min before being placed in culture for the indicated time. In the kynurenine uptake, Myc-GFP and Myc staining assays the naïve control cells were cultured in 5 ng/ mL IL-7 (Peprotech). For measurement of the effect of mTORC1 inhibition on Myc-GFP expression (*Figure 4E*) Myc-GFP$^{KI}$ T cells were cultured +/- rapamycin (20 nM, Merck) for the indicated time. For measurement of Myc-GFP in amino acid deficient conditions (*Figure 4F–G*) Myc-GFP$^{KI}$ T cells were cultured in RPMI 1640 (+aa), RPMI – Leucine (-Leu, Sigma), RPMI – Methionine (-Met, DC Biosciences), RPMI – Glutamine (-Gln, Gibco) or HBSS (-aa, Gibco) supplemented with 10% dialyzed FBS (Gibco), penicillin/streptomycin and 50 µM β-mercaptoethanol. For Myc staining in *Figure 1D* splenic single cell suspensions were activated with 0.5 µg/mL of both anti-CD3 and anti-CD28 at a density of 1 million live cells/mL. To determine system L uptake and Myc co-expression, Myc-GFP$^{KI}$ (CD45.2) and WT (CD45.1, to provide an internal control for autofluorescence) splenocytes were mixed together at a ratio of 1:1 prior to activation.

For CFSE assay, lymph node suspensions at 1 million cells per mL in PBS 1%FCS were labelled with 5 µM CFSE (Invitrogen) for 10 min at 37°C before being washed twice with cold complete medium to quench the reaction. 0.5 million live cells per mL were activated with 0.5 µg/mL of both anti-CD3 + anti-CD28.

For IFNγ and Granzyme B intracellular staining, splenocytes were activated with anti-CD3 and anti-CD28 (both 0.5 µg/mL) for 20 hr at a density of 1 million cells/mL. Golgi plug (1:1000, BD Biosciences) +/- PdBU (20 ng/mL) and ionomycin (500 ng/mL) were added to the culture for 4 hr then cells were harvested after 24 hr total time in culture.

## Cell sorting

Cell sorting was performed on a BD Influx cell sorter. Staining, sorting and cell collection was performed in RPMI 1640 containing glutamine, supplemented with 1% FBS.

Naïve CD4 and CD8 T cells for proteomics and qPCR were sorted from lymph nodes of 2 x Cd4*Cre*$^+$ mice per biological replicate. Naïve cells were sorted from single cell suspensions as DAPI$^-$B220$^-$NK1.1$^-$CD11b$^-$CD25$^-$TCRb$^+$CD62L$^{hi}$CD44$^{lo}$, CD4$^+$ or CD8$^+$.

24 hr activated T cells for Myc$^{WT}$ and Myc$^{cKO}$ proteomics were sorted as DAPI$^-$CD69$^+$ CD4$^+$ or CD8$^+$.

24 hr activated T cells for Slc7a5$^{WT}$ and Slc7a5$^{cKO}$ proteomics were sorted as Slc7a5$^{WT}$: DAPI$^-$CD4$^+$CD45.1$^+$CD45.2$^-$; and Slc7a5$^{cKO}$: DAPI$^-$CD4$^+$CD45.1$^-$CD45.2$^+$.

4 hr activated T cells for qPCR were sorted as DAPI$^-$B220$^-$NK1.1$^-$CD11b$^-$Thy1.2$^+$CD69$^+$ CD4$^+$ or CD8$^+$.

In all cases, sorted cells were washed twice with HBSS before being snap frozen in liquid nitrogen and stored at −80°C until further processing.

## Flow cytometry

Flow cytometry data was acquired on a FACSVerse using FACSuite software or FACSCanto, or LSR II Fortessa with FACS DIVA software (BD Biosciences). Data was analysed using Flowjo software version 9.9.6 (Treestar).

For cell surface staining antibodies conjugated to BV421, BV510, FITC, PE, PerCPCy5.5, PECy7, APC, and APCeF780 were obtained from BD Biosciences, eBioscience or Biolegend. Fc receptors were blocked using Fc block (BD Biosciences). Antibody clones were as follows: CD4 (RM4-5), CD8 (53–6.7), CD11b (M1/70), CD25 (7D4), CD44 (IM7), CD45.1 (A20), CD45.2 (104), CD62L (MEL-14), CD69 (H1.2F), TCRbeta (H57-597), Thy1.2 (53–2.1), B220 (RA3-6B2), NK1.1 (PK136).

For Myc intracellular staining, cells were fixed and permeabilised overnight in PBS 1% FBS 0.5% PFA 0.2% Tween-20. Fix/perm was washed off and cells were stained with 1:200 rabbit anti-Myc antibody (Cell Signalling Technologies, clone D84C12, cat#5605S) for 1 hr at room temperature followed by 1:1000 anti-rabbit IgG (H+L) F(ab')$_2$ AlexFluor647 secondary antibody (Cell Signalling Technologies, cat#4414S) for 1 hr at room temperature.

For IFNγ and Granzyme B intracellular staining cells were fixed and permeabilised using eBioscience Intracellular Fixation and Permeabilisation kit (eBioscience) as per manufacturer instructions. Cells were stained with anti-IFNγ (XMG1.2) and anti-Granzyme B (NGZB) at 1:100 and 1:200 respectively.

## Kynurenine uptakes

Kynurenine uptake assay were performed as described in *Sinclair et al. (2018)*. Briefly, antibodies against surface markers were added to culture (37°C 5% CO$_2$) for 10 min prior to uptake assay in order to identify cell types. ~ 3 million cells per condition were harvested, washed with warm HBSS and split into three wells/tubes in warm HBSS. For each condition either HBSS, Kynurenine (200 μM final concentration) or BCH (10 mM, a system L inhibitor) + Kynurenine were added and placed back at 37°C. Uptakes were stopped after 5 min by addition of PFA (final concentration 1%) for 30 min at room temp. After fixation cells were washed with PBS 1%FBS before analysis on flow cytometer. Kynurenine is excited by the 405 nm laser and is detected in the 450/50 BP filter.

## qPCR

Total RNA was isolated from sorted pellets using RNeasy Minikit (Qigen) with on column DNase (Qiagen) digestion and cDNA transcribed using iScript cDNA Synthesis kit (Biorad, cat#1708891) all as per manufacturer instructions. Quantitative Real-Time PCR was performed using iTaq Universal SYBRGreen Supermix (Biorad, cat#1725121) on a Bio-Rad iQ5 Multicolor Real-Time PCR Detection System, with Bio-Rad iQ5 software. mRNA fold-change was quantified relative to naïve CD4 Myc$^{WT}$ T cells using the ΔΔCt method, with *TBP* as the loading control. Primer sequences were as follows:

Slc7a5:

Forward; AAG GCT GCG ACC CGT GTG
Reverse; ATC ACC TTG TCC CAT GTC CTT CC

Slc7a1:

Forward; GGA GCT TTG GC CTT CAT CAC T
Reverse; CAG CAC CCC AGG AGC ATT CA

Slc1a5:

Forward; GCC ATC ACC TCC ATC AAC GAC T
Reverse; AGA GCG GAA GGC AGC AGA CAC

TBP:

Forward: GTG AAT CTT GGC TGT AAA CTT GAC CT
Reverse: CGC AGT TGT CCG TGG CTC T

## Proteomics sample preparation

Cell pellets were lysed at room temperature in 4% SDS, 50 mM TEAB pH 8.5, 10 mM TCEP under agitation (5 min, 1200 rpm on tube shaker), boiled (5 min, 500 rpm on tube shaker), then sonicated with a BioRuptor (30 s on, 30 s off x30 cycles). Protein concentration was determined using EZQ protein quantitation kit (Invitrogen) as per manufacturer instructions. Lysates were alkylated with 20 mM iodoacetamide for 1 hr at room temperature in the dark, before protein clean up by SP3 procedure (*Hughes et al., 2014*). Briefly, 200 μg of 1:1 mixed Hydrophobic and Hydrophilic Sera-Mag Speed-Bead Carboxylate-Modified Magnetic Particles were added per protein sample then acidified to ~pH 2.0 by addition 10:1 Acetonitrile: Formic Acid. Beads were immobilised on a magnetic rack and proteins washed with 2 × 70% ethanol and 1 × 100% acetonitrile. Rinsed beads were reconstituted in 0.1% SDS 50 mM TEAB pH 8.5, 1 mM CaCl2 and digested overnight with LysC followed by overnight digestion with Trypsin, each at a 1:50 enzyme to protein ratio. Peptide clean up was

performed as per SP3 procedure (*Hughes et al., 2014*). Briefly, protein-bead mixtures were resuspended and 100% acetonitrile added for 10 min (for the last 2 min of this beads were immobilised on a magnetic rack). Acetonitrile and digest buffer were removed, peptides were washed with acetonitrile and eluted in 2% DMSO. Peptide concentration was quantified using CBQCA protein quantitation kit (Invitrogen) as per manufacturer protocol. Formic acid was added to 5% final concentration.

Samples were fractionated using high pH reverse phase liquid chromatography. Samples were loaded onto a 2.1 mm x 150 mm XBridge Peptide BEH C18 column with 3.5 μm particles (Waters). Using a Dionex Ultimate3000 system, the samples were separated using a 25 min multistep gradient of solvents A (10 mM formate at pH 9 in 2% acetonitrile) and B (10 mM ammonium formate pH 9 in 80% acetonitrile), at a flow rate of 0.3 mL/min. Peptides were separated into 16 fractions which were consolidated into eight fractions. Fractionated peptides were dried in vacuo then dissolved in 5% Formic Acid for analysis by LC-ES-MS/MS. For Myc$^{WT}$ naïve and OT1 TCR time course proteomics samples clean up was performed on the 8$^{th}$ fraction of each sample using HIPPR detergent removal spin column kit (ThermoFisher Scientific) as per manufacturer protocol.

## Liquid chromatography electrospray tandem mass spectrometry analysis (LC-ES-MS/MS)

≤1 μg of peptide was analysed per fraction in all experiments.

For label-free proteomics of Myc$^{WT}$ and Myc$^{cKO}$ and Slc7a5$^{WT}$ and Slc7a5$^{cKO}$ 24 hr TCR activated T cells samples were analysed as described previously (*Sinclair et al., 2019*). As described in reference, samples were injected onto a nanoscale C18 reverse-phase chromatography system (UltiMate 3000 RSLC nano, Thermo Scientific) then electrosprayed into an Orbitrap mass spectrometer (LTQ Orbitrap Velos Pro; Thermo Scientific). For chromatography buffers were as follows: HPLC buffer A (0.1% formic acid), HPLC buffer B (80% acetonitrile and 0.08% formic acid) and HPLC buffer C (0.1% formic acid). Peptides were loaded onto an Acclaim PepMap100 nanoViper C18 trap column (100 μm inner diameter, 2 cm; Thermo Scientific) in HPLC buffer C with a constant flow of 10 μl/min. After trap enrichment, peptides were eluted onto an EASY-Spray PepMap RSLC nanoViper, C18, 2 μm, 100 Å column (75 μm, 50 cm; Thermo Scientific) using the buffer gradient: 2% B (0 to 6 min), 2% to 35% B (6 to 130 min), 35% to 98% B (130 to 132 min), 98% B (132 to 152 min), 98% to 2% B (152 to 153 min), and equilibrated in 2% B (153 to 170 min) at a flow rate of 0.3 μl/min. The eluting peptide solution was automatically electrosprayed using an EASY-Spray nanoelectrospray ion source at 50° and a source voltage of 1.9 kV (Thermo Scientific) into the Orbitrap mass spectrometer (LTQ Orbitrap Velos Pro; Thermo Scientific). The mass spectrometer was operated in positive ion mode. Full-scan MS survey spectra (mass/charge ratio, 335 to 1800) in profile mode were acquired in the Orbitrap with a resolution of 60,000. Data were collected using data- dependent acquisition: the 15 most intense peptide ions from the preview scan in the Orbitrap were fragmented by collision-induced dissociation (normalized collision energy, 35%; activation Q, 0.250; activation time, 10 ms) in the LTQ after the accumulation of 5000 ions. Precursor ion charge state screening was enabled, and all unassigned charge states as well as singly charged species were rejected. The lock mass option was enabled for survey scans to improve mass accuracy. (Using Lock Mass of 445.120024).

For label-free proteomics of naïve WT T cells and N4 activated OT-I T cell time course, samples were injected onto a nanoscale C18 reverse-phase chromatography system (UltiMate 3000 RSLC nano, Thermo Scientific) before being electrosprayed into a Q Exactive Plus mass spectrometer (Thermo Scientific). The chromatography buffers used were as follows: HPLC buffer A (0.1% formic acid), HPLC buffer B (80% acetonitrile in 0.1% formic acid) and HPLC buffer C (0.1% formic acid). Samples (15 μL) were injected and washed with Buffer C (10 ul/min) for 5 min prior to valve switch on an Acclaim PepMap100 nanoViper C18 trap column (100 μm inner diameter, 2 cm; Thermo Scientific). After trap enrichment, peptides were eluted onto an EASY-Spray PepMap RSLC nanoViper, C18, 2 μm, 100 Å column (75 μm, 50 cm; Thermo Scientific) using the following buffer gradient: 2% to 5% B (0 to 5 min), 5% to 35% B (5 to 130 min), 35% to 98% B (130 to 132 min), 98% B (132 to 152 min), 98% to 2% B (152 to 153 min), and equilibrated in 2% B (153 to 170 min) at a flow rate of 0.3 μl/min. The eluting peptide solution was automatically electrosprayed into the Q Exactive Plus mass spectrometer using an EASY-Spray nanoelectrospray ion source at 50° and a source voltage of 2.0 kV (Thermo Scientific). The mass spectrometer was operated in positive ion mode. Data were

collected using data-dependent acquisition: the 15 most intense peptide ions from the preview scan in the Q Exactive Plus were fragmented by higher-energy collisional dissociation. The following settings were applied: MS1 scan resolution: 70 000; MS1 AGC target: 1e6; MS1 maximum IT: 20 ms; MS1 scan range: 350–1600 Th; MS2 scan resolution: 17 500; MS2 AGC target: 2e5; MS2 maximum IT: 100 ms; isolation window: 1.4 Th; first fixed mass: 200 Th; NCE: 27; minimum AGC target: 2e3; only charge states 2 to 6 considered; peptide match: preferred; exclude isotopes: on; dynamic exclusion: 45 s.

## Quantification and statistical analysis

### Proteomics data analysis

The data were processed, searched and quantified with the MaxQuant software package, Version 1.6.2.6. For the protein and peptide searches we generated a hybrid database from databases in Uniprot release 2019 07. This consisted of all manually annotated mouse SwissProt entries, combined with mouse TrEMBL entries with protein level evidence available and a manually annotated homologue within the human SwissProt database. The following MaxQuant search parameters were used: protein N-terminal acetylation, methionine oxidation, glutamine to pyroglutamate, and glutamine and asparagine deamidation were set as variable modifications and carbamidomethylation of cysteine residues was selected as a fixed modification; Trypsin and LysC were selected as the enzymes with up to two missed cleavages permitted; the protein and PSM false discovery rate was set to 1%; matching of peptides between runs was switched off. Data filtering and protein copy number quantification was performed in the Perseus software package, version 1.6.6.0. Proteins were quantified from unique peptides and razor peptides (peptides assigned to a group, but not unique to that group). Quantification quality was categorized based on the following: quantification was considered high accuracy if proteins had eight or more unique and razor peptides assigned and at least 75% of these peptides were unique; proteins were considered medium accuracy if they were assigned at least three unique and razor peptides with 50% of these being unique; proteins below these thresholds were considered low accuracy. The data set was filtered to remove proteins categorised as 'contaminants', 'reverse' and 'only identified by site'. Mean copy number per cell was calculated using the "proteomic ruler' plugin as described in *Wiśniewski et al. (2014)*. Briefly, this method sets the summed peptide intensities of the histones to the number of histones in a diploid mouse cell then uses the ratio between the histone peptide intensity and summed peptide intensities of other identified proteins to estimate the protein copy number per cell for all the identified proteins. Data was further filtered to only include proteins for which at least one condition had peptides detected in ≥2 biological replicates.

### Statistics and calculations

Mass contribution of proteins (g/cell) was calculated as (protein copy number) * (molecular weight (Daltons)) / (Avogadro's constant). Protein content per cell plots for glycolytic enzymes in *Figure 2C* were calculated based upon proteins defined as: KEGG term 'glycolysis + gluconeogenesis' manually filtered to exclude enzymes not directly part of the glycolysis pathway illustrated in *Figure 2A*.

Heatmaps were generated using Broad Institute software Morpheus (https://software.broadinstitute.org/morpheus). Proteins were included in heatmap if they had an average of at least 500 copies per cell and were detected in at least two biological replicates in the CD8$^+$ T cell conditions.

P-values were calculated using two-tailed t-test with unequal variance on log2 transformed copy number per cell values in Microsoft Excel. Mean fold-changes between average copy number of conditions were calculated. For *Figure 3J–K*, proteins were considered TCR regulated if the naïve WT to TCR WT p-value was <0.05 irrespective of fold-change; proteins were considered Myc or Slc7a5 regulated if they differed by >2 fold between WT TCR and cKO TCR conditions with a p-value<0.05. For *Figure 1—figure supplement 2* proteins were considered Myc-regulated if they differed by >2 fold between Myc$^{WT}$ TCR and Myc$^{cKO}$ TCR conditions with a p-value<0.05.

## Acknowledgements

The authors thank members of the Cantrell group for their critical discussion of the data. We thank A Richards and G Griffith for providing processed single cell RNAseq data and for critical comments

on the manuscript. We thank B Stubbs and E Emslie for technical assistance with mouse genotyping and qPCR experiments, A Brenes for assistance with generating hybrid proteomic search database, T Youdale and C Rollings for assistance with generating samples for OT1 TCR timecourse proteomics, A Gardner, A Rennie and R Clarke from the Flow Cytometry facility for cell sorting, K Beattie from the Fingerprint proteomics facility for the mass spectrometry and advice on proteomics and the Biological Resource unit at the University of Dundee. This research was supported by a Wellcome Trust Principal Research Fellowship to DAC (205023/Z/16/Z), a Wellcome Trust Equipment Award to DAC (202950/Z/16/Z), an EMBO Long term Fellowship to JMM (ALTF 1543–2015) and has received funding from the European Union's Horizon 2020 research and innovation programme under the Marie Skłodowska-Curie grant agreement No 705984 to JMM and DAC.

## Additional information

### Funding

| Funder | Grant reference number | Author |
|---|---|---|
| Wellcome | 097418/Z/11/Z | Doreen A Cantrell |
| Wellcome | 205023/Z/16/Z | Doreen A Cantrell |
| Wellcome | 202950/Z/16/Z | Doreen A Cantrell |
| European Molecular Biology Organization | ALTF 1543-2015 | Julia M Marchingo |
| H2020 Marie Skłodowska-Curie Actions | 705984 | Julia M Marchingo Doreen A Cantrell |

The funders had no role in study design, data collection and interpretation, or the decision to submit the work for publication.

### Author contributions

Julia M Marchingo, Conceptualization, Formal analysis, Investigation, Visualization, Methodology, Writing - original draft, Writing - review and editing; Linda V Sinclair, Conceptualization, Investigation, Visualization, Methodology, Writing - original draft, Writing - review and editing; Andrew JM Howden, Formal analysis, Investigation, Methodology, Writing - review and editing; Doreen A Cantrell, Conceptualization, Supervision, Funding acquisition, Investigation, Visualization, Writing - original draft, Project administration, Writing - review and editing

### Author ORCIDs

Julia M Marchingo ⓘD https://orcid.org/0000-0001-8823-9718
Linda V Sinclair ⓘD https://orcid.org/0000-0003-1248-7189
Andrew JM Howden ⓘD http://orcid.org/0000-0002-4332-9469
Doreen A Cantrell ⓘD https://orcid.org/0000-0001-7525-3350

### Ethics

Animal experimentation: All animal experiments were performed under Project License PPL 60/4488 and P4BD0CE74. The University of Dundee Welfare and Ethical Use of Animals Committee accepted the project license for submission to the Home Office. Mice were bred and maintained in the WTB/RUTG, University of Dundee in compliance with UK Home Office Animals (Scientific Procedures) Act 1986 guidelines.

### Decision letter and Author response

Decision letter https://doi.org/10.7554/eLife.53725.sa1
Author response https://doi.org/10.7554/eLife.53725.sa2

## Additional files

### Supplementary files
• Supplementary file 1. Copy number, protein content, fold-change calculations, statistical tests and heatmap input from proteomics data.

• Transparent reporting form

### Data availability
All data generated or analysed during this study are included in the manuscript and supporting files. Raw mass spec data files and MaxQuant analysis files for naïve WT, and TCR activated MycWT, MyccKO, Slc7a5WT and Slc7a5cKO T cells are available on the ProteomeXchange data repository and can be accessed with identifier PXD016105 (https://www.ebi.ac.uk/pride/archive/projects/PXD016105). Raw mass spec data files and MaxQuant analysis files for OT-1 TCR time-course data are available on the ProteomeXchange data repository and can be accessed with identifier PXD016443 (https://www.ebi.ac.uk/pride/archive/projects/PXD016443).

The following datasets were generated:

| Author(s) | Year | Dataset title | Dataset URL | Database and Identifier |
|---|---|---|---|---|
| Marchingo JM, Sinclair LV, Howden AJM, Cantrell DA | 2019 | Proteome of naive and TCR activated wild-type, Myc-deficient and Slc7a5-deficient T cells | https://www.ebi.ac.uk/pride/archive/projects/PXD016105 | PRIDE, PXD016105 |
| Marchingo JM, Sinclair LV, Howden AJM, Cantrell DA | 2019 | OT1 T cell activation time course | https://www.ebi.ac.uk/pride/archive/projects/PXD016443 | PRIDE, PXD016443 |

The following previously published dataset was used:

| Author(s) | Year | Dataset title | Dataset URL | Database and Identifier |
|---|---|---|---|---|
| Richard AC, Lun ATL, Lau WWY, Gottgens B, Marioni JC, Griffiths GM | 2018 | Single-cell RNA sequencing of OT-I CD8+ T cells after stimulation with different affinity ligands | https://www.ebi.ac.uk/arrayexpress/experiments/E-MTAB-6051/ | ArrayExpress, E-MTAB-6051 |

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
