## [Decision Letter]

**Acceptance summary:**

This study makes important contributions to our understanding of Myc in primary T cells and the data support a novel, essential gatekeeper function for Myc to accumulate cell mass when T cells are stimulated. It also provides a clear example of how quantitative and temporally resolved proteomics can provide fundamental insights into biological processes.

**Decision letter after peer review:**

Thank you for submitting your article "Quantitative analysis of how Myc controls T cell proteomes and metabolic pathways during T cell activation" for consideration by *eLife*. Your article has been reviewed by three peer reviewers, including Ellen A Robey as the Reviewing Editor and Reviewer #1, and the evaluation has been overseen by a Reviewing Editor and Tadatsugu Taniguchi as the Senior Editor. The following individuals involved in review of your submission have agreed to reveal their identity: Jeroen P Roose (Reviewer #3).

The reviewers have discussed the reviews with one another and the Reviewing Editor has drafted this decision to help you prepare a revised submission.

Summary:

The study by Marchingo et al., presents several important findings. The authors compared the proteomes of control and Myc deficient T cells to investigate how Myc controls early T cell activation events. They identified a role for Myc in selectively modulating the expression of various T cell activation and metabolic pathways. The authors identify Myc dependent upregulation of amino acid transporters such as Slc7a5 as a key checkpoint controlling T cell growth. In contrast to earlier work by the same group showing that protein expression of Myc is dependent on Slc7a5 expression, the authors demonstrate that early Myc upregulation is Slc7a5 independent and likely contributes to feedforward maintenance of amino acid transport.

Essential revisions:

1) The main question the authors addressed is how Myc controls antigen receptor driven cell growth. However, 24 hours of activation also induces IL-2/IL-2R signaling. The authors should comment on the contribution of IL-2/IL-2R signaling to the results they observe, particularly as IL-2 signaling in control T cells are likely to amplify Myc dependent proteome restructuring. This could also be put into context with earlier work by the same group showing that both antigen and IL2 can regulate amino acid transport.

2) The authors use several different stimulation conditions – anti-CD3/28 at 0.5µg/mL or 3µg/mL or OVA peptide at 10ng/mL (+ hIL-2 in the case of Slc7a5 KO T cells) for their proteome analyses. Could the authors provide some information that these different conditions are roughly equivalent in terms of the proteins/pathways they engage?

3) The authors compared the proteomes of control and Myc KO in both CD4 and CD8 T cells. Are there any differences in how Myc selectively regulates protein expression in CD4 versus CD8 T cells?

4) The authors demonstrate strong overlap of Myc KO and Slc7a5 KO proteomes; what are their differences?

5) The finding that early Myc expression is Slc7a5 independent is interesting and correlates nicely with induction of Myc occurring prior to Slc7a5. Is early Myc expression dependent on amino acid uptake to any extent? For example, would activating T cells in the absence of key amino acids impair early Myc upregulation?

6) Figure 1H to 1O should have statistical analysis. The same holds for the bar graphs in the other figures.

7) In Figure 2, the authors explore glycolytic metabolism and glutamine catabolism. This is important because other work has made links between Myc and these processes. It would be valuable to include some functional assays here. For example, are Myc KO cells more sensitive to pharmacological inhibition of these processes?

8) It would be valuable to have a more extensive time course of the data in Figure 3E, as this would nicely connect to the panels presented in Figure 4.

9) It would be informative for the reader to connect the data presented in Figure 4D to their other studies. There is some text in the Discussion section on mTOR but an experiment would help here. Is the initial induction of Myc independent of mTOR and independent of Slc7a5? Is the sustained induction of Myc independent of mTOR but dependent of Slc7a5? Or not? Either outcome is fine but seeing data with an mTOR inhibitor in Figure 4D would be informative.

---

## [Author Response]

Essential revisions:1) The main question the authors addressed is how Myc controls antigen receptor driven cell growth. However, 24 hours of activation also induces IL-2/IL-2R signaling. The authors should comment on the contribution of IL-2/IL-2R signaling to the results they observe, particularly as IL-2 signaling in control T cells are likely to amplify Myc dependent proteome restructuring. This could also be put into context with earlier work by the same group showing that both antigen and IL2 can regulate amino acid transport.

This is a very good question as autocrine signalling by IL-2 is important for T cells.

In this context, under the conditions that we use in the present study, the impact of autocrine secreted IL-2 in the cultures is relatively minimal at 24 hours. This conclusion is based on experiments where we have looked at the effect of blocking IL-2 with the anti-IL-2ra antibody clone PC61. Our results indicate that blocking IL-2 signalling has minimal impact on Myc expression, CD71 expression (a downstream target of Myc), CD98 expression (the heavy chain of the System L amino acid transporters) or cell size at a 24 hour time point after TCR stimulation but has very strong effects at a 48 hour time point. It is also relevant that Myc null T cells make IL-2 in response to TCR activation (Wang, R et al., 2011, supp Figure 4) as do Slc7a5 null T cells (Sinclair, LV, 2013, Figure 6). As such, we think that any effects of IL-2/IL2R signalling on Myc driven proteome remodelling are negligible in the current study. They would be highly relevant if we looked at any later time points.

We have modified the text to reflect this in subsection “Selective remodelling of T cell proteomes by Myc”

“To examine how Myc loss impacts proteome remodelling during immune activation we performed quantitative label-free high-resolution mass spectrometry on 24 hour CD3/CD28 activated wild-type (CD4cre^+^, Myc^WT^) and Myc^cKO^ CD4^+^ and CD8^+^ T cells. This time point was chosen as it is when we observe maximal increase in cell size of the immune activated cells with no difference in survival between Myc^WT^ and Myc^cKO^ T cells. Moreover, at this time point there is minimal impact of autocrine secreted cytokine IL-2 on Myc expression (Figure 1—figure supplement 1B).”

2) The authors use several different stimulation conditions – anti-CD3/28 at 0.5µg/mL or 3µg/mL or OVA peptide at 10ng/mL (+ hIL-2 in the case of Slc7a5 KO T cells) for their proteome analyses. Could the authors provide some information that these different conditions are roughly equivalent in terms of the proteins/pathways they engage?

In our experience these stimuli have broadly similar effects when one is looking at cells activated at a 24 time point. Where they do differ is in the kinetics of the response they induce; if one was looking at early time points then it is well established that the signal strength will drive 'digitality' in T cell responses. However, we find that by the time one gets to a 24 hour time point, the cells have all responded and are very similar in terms of expression of the key transcription factors, cytokine receptors and the key proteins for T cell metabolism etc. It is a very intriguing question (beyond the scope of the present study) as to whether the signalling pathways used by CD3/CD28 antibodies versus peptides are all the same? We know that in the mouse when one uses aCD3 (2C11) and aCD28 (37.51) or peptide MHC complexes then these signals all converge on Myc. We have also compared how inhibition of mTORC1, ERK signalling, PI3K signalling, Protein Kinase D signalling impact on T cells activated with CD3/CD28 antibodies versus peptide and to date have found minimal differences. We would stress that we would only be certain about this point with mouse T cells and the different conditions used in the current study. Many years ago, using human T cells and antibodies to human receptors, the Cantrell group showed that different CD28 antibodies could differentially activate Ras or PI3K. Furthermore, some antibodies against human CD3 antigens are excellent activators of PI3K (UCHT1), some are not (OKT3).

3) The authors compared the proteomes of control and Myc KO in both CD4 and CD8 T cells. Are there any differences in how Myc selectively regulates protein expression in CD4 versus CD8 T cells?

We had not directly addressed this question in our original manuscript and our answer is that Myc has a qualitatively similar effect on CD4 and CD8 T cell proteome remodelling. However, there are quantitative differences in the effects of Myc which reflect that the antigen receptor and Myc driven biomass expansion is larger in CD8 than in CD4s i.e. the higher they rise the further they fall.

To illustrate and discuss this aspect of the data we have created an additional supplementary figure (Figure 3—figure supplement 2) and added the following text to subsection “Selective remodelling of T cell proteomes by Myc”:

“The selective effects of Myc-deficiency on protein expression in activated CD8^+^ and CD4^+^ T cells appeared qualitatively similar (Figure 1G). There were however some quantitative differences. These differences reflect that some proteins were more highly expressed in activated Myc^WT^ CD8^+^ T cells than in Myc^WT^ CD4^+^ T cells, however, Myc-deficiency reduced protein expression down to a similar level in both CD4^+^ and CD8^+^ T cells, therefore giving a larger effect size in CD8^+^ T cells (Figure 1—figure supplement 2). When taken in conjunction with the observation that CD8^+^ T cells expressed a higher level of Myc (Figure 1C-D), associated with increased cell biomass (Figure 1A-B), this suggests a dose-dependent Myc-driven amplification of protein expression.

Collectively, these data show that immune activated T cell proteome remodelling comprises both Myc dependent and independent processes and that Myc has a qualitatively similar, but dose-dependent effect on CD4^+^ and CD8^+^ T cell proteomes.”

4) The authors demonstrate strong overlap of Myc KO and Slc7a5 KO proteomes; what are their differences?

This is an interesting question and one that we had not dealt with in the original manuscript due to fear of making the paper too unfocused. It is perhaps pertinent to say that we were actually amazed at the major impact that losing a single amino transporter had on the proteome of activated T cells. The answer is that there are some differences in between Slc7a5 null T cells and Myc null T cells; Slc7a5 null T cells have zero System L transport capacity whereas the Myc null T cells have low basal System L transport capacity in naive cells but fail to upregulate amino acid transport in response to immune stimulation.

To discuss this, we have modified the manuscript and included examples of where protein expression differs between Myc KO and Slc7a5 KO T cells, including proteomics data of the glucose transporter Slc2a3, and flow cytometry data for IFNgamma and Granzyme B. The following text has been added to subsection “Myc controls amino acid transporter expression in immune activated T cells”:

“Although there is a large degree of overlap in the proteomics data, Slc7a5-deficiency does not completely phenocopy the effects of Myc-deficiency. Induction of proteins such as the glucose transporter Slc2a3 (Figure 2B, Figure 3—figure supplement 4C) and effector molecules like Granzyme B and IFNγ (Figure 3—figure supplement 4D-E) exhibit a more severe defect in Slc7a5^cKO^ T cells. This is likely due to the lack of basal-level amino acid transport in naïve Slc7a5^cKO^ T cells which is not deficient in naïve Myc^cKO^ T cells (Figure 3C).”

Note all the proteomics data in this paper will be freely available for others to explore and make comparisons.

5) The finding that early Myc expression is Slc7a5 independent is interesting and correlates nicely with induction of Myc occurring prior to Slc7a5. Is early Myc expression dependent on amino acid uptake to any extent? For example, would activating T cells in the absence of key amino acids impair early Myc upregulation?

This question is very interesting, so we performed more experiments and have now included additional data that addresses this question. New panels in Figure 4F and G show the impact of withdrawal of all amino acids or the individual amino acids Glutamine, Methionine and Leucine on TCR induction of Myc expression. The key result is that, yes, the induction of Myc does require exogenous amino acids. It’s a bit convoluted; completely amino acid free media cannot support Myc induction (Figure 4F) however, deficiency of a single amino acid from the media, such as glutamine, methionine or leucine, leads to a reduction in Myc levels but does not prevent its expression (Figure 4G).

We have added new figures and additions to the main text and methods to describe and comment upon these data (see subsection “Myc induction of amino acid transport initiates a positive feedforward loop”).

“Although Slc7a5 deficiency or mTORC1 inhibition alone was insufficient to prevent Myc induction, the presence of external amino acids is necessary for expression of Myc protein, with T cells activated in amino acid-free media being unable to express Myc protein despite increasing the activation marker CD69 (Figure 4F). Deficiency of a single amino acid from the media, such as glutamine, methionine or leucine leads to a reduction in Myc levels but does not prevent its expression (Figure 4G).”

6) Figure 1H to 1O should have statistical analysis. The same holds for the bar graphs in the other figures.

In all bar graphs of proteomics data, we plot data from each biological replicate. We have not placed lines/stars indicating statistical tests above the figures as we feel this unnecessarily clutters the figure with a large number of comparisons where the differences that we highlight in the main text are visually quite clear. However, the supplementary table does contain p values and fold-changes for all proteins, including statistical testing from comparisons made within the main text under the worksheet tab “fold-change and t test”. Note, since the data were log-transformed fold-changes/p-values for proteins where multiple copy numbers for a given condition were = 0 was not possible. We have now highlighted the availability of results from statistical testing in the figure legends for anyone wishing to look up this information for a particular comparison/protein by including the statement…

“Fold-change calculations and statistical testing comparing naïve WT vs TCR Myc^WT^, naïve WT vs TCR Myc^cKO^, and TCR Myc^WT^ vs TCR Myc^cKO^ protein copy number per cell is listed in Supplementary file 1.”

7) In Figure 2, the authors explore glycolytic metabolism and glutamine catabolism. This is important because other work has made links between Myc and these processes. It would be valuable to include some functional assays here. For example, are Myc KO cells more sensitive to pharmacological inhibition of these processes?

With regard to experiments with pharmacological inhibitors, we could not think of specific tools. Many people use tools to target respiratory chain complexes or target glycolysis, however these usually activate AMPK and inhibit mTORC1 thus making it hard to interpret experiments. Moreover, the metabolism of Myc null T cells is already so compromised and we were not able to generate a specific hypothesis for an experiment that would give us new insight into T cell metabolic pathway regulation in these cells.

What we did discuss, but possibly with insufficient emphasis, is that Wang and colleagues (Wang et al., 2011) have used both label free and targeted (tracing) metabolomics to show that Myc null T cells have defective flux through glycolysis and glutaminolysis. We have also shown that Myc null cells have defective o-GlcNACylation (Swamy et al., 2016). The new insight from the current study is protein-level information about why Myc cells may have these defects.

Furthermore, we have interrogated the proteomics data to determine key protein expression of other metabolic pathways in T cells, eg branched chain amino acid metabolism and methionine metabolism (Figure 3—figure supplement 3). Protein expression levels of these pathways is a valuable addition to our understanding of what other metabolic pathways might play a role in T cell activation.

8) It would be valuable to have a more extensive time course of the data in Figure 3E, as this would nicely connect to the panels presented in Figure 4.

Unfortunately, we did not do a matched System transport assay in parallel with the proteomics data shown in Figure 4. However, the key question being asked by the reviewer seems to be if amino acid transport activity correlates with transporter numbers. We now present System L transport data from later time points (6 hours and 24 hours) which are covered by the proteomics experiment in Figure 3—figure supplement 2. These data show that System L transport increases substantially between 6 hours and 24 hours after TCR activation in Myc^WT^ T cells compared to naïve (IL7 maintained cells) but fails to do so in Myc^cKO^ T cells. The proteomics data show a clear increase in Slc7a5 protein expression from a basal level in naïve cells, at 6 hours and further increased at 24 hours in Figure 4B. Moreover, activated CD4 T cells have lower Slc7a5 copy numbers than activated CD8 cells and corresponding differences in System L transport. Collectively this gives us confidence that the only thing that limits transport is transporter numbers.

We have added the following text to subsection “Myc controls amino acid transporter expression in immune activated T cells”:

“There was also a strong correlation between the levels of Myc protein expressed by activated T cells and system L amino acid transport capacity (Figure 3F) and while system L transport continued to increase over the first 24 hours of T cell activation in Myc^WT^ T cells this did not occur in Myc^cKO^ T cells (Figure 3—figure supplement 2A-B). “

And in subsection “Myc controls amino acid transporter expression in immune activated T cells”:

“Proteomics data shows that expression of amino acid transporters increases gradually over time (Figure 4B), and Kyn uptake experiments confirm that this increase in transporter number corresponds with higher system L uptake (Figure 3—figure supplement 2).”

9) It would be informative for the reader to connect the data presented in Figure 4D to their other studies. There is some text in the Discussion section on mTOR but an experiment would help here. Is the initial induction of Myc independent of mTOR and independent of Slc7a5? Is the sustained induction of Myc independent of mTOR but dependent of Slc7a5? Or not? Either outcome is fine but seeing data with an mTOR inhibitor in Figure 4D would be informative.

We have now included an additional experiment in Figure 4E that addresses this question by measuring Myc expression after 4 and 24 hours TCR activation +/- the mTORC1 inhibitor rapamycin. We have modified the text to describe these data as follows in subsection “Myc controls amino acid transporter expression in immune activated T cells”:

“The finding that Slc7a5^cKO^ T cells can induce Myc expression is also surprising in the context of previous work demonstrating an important role for mTORC1 activation (which is critically dependent on uptake of leucine) in controlling Myc expression during T cell activation (Wang et al., 2011). Therefore, we tested the dependency of Myc protein expression on mTORC1 signalling in our T cell system. The data show that rapamycin treatment had very little impact on expression of Myc protein after 4 hours of T cell activation but did reduce Myc expression after 24 hours (Figure 4E), consistent with the results seen in Slc7a5^cKO^ T cells (Figure 4D).”